# PLanTS: Periodicity-aware Latent-state Representation Learning for Multivariate Time Series

## Abstract

Multivariate time series (MTS) data are ubiquitous in domains such as healthcare, climate science, and industrial monitoring, but their high dimensionality, scarce labels, and non-stationary nature pose significant challenges for conventional machine learning methods. While recent self-supervised learning (SSL) approaches mitigate label scarcity by data augmentations or time point-based contrastive strategy, they overlook the intrinsic periodic structure of MTS and fail to capture the dynamic evolution of latent states. We propose PLanTS, a periodicity-aware self-supervised learning framework that explicitly models irregular latent states and their transitions. We first designed a periodicity-aware multi-granularity patching mechanism and a generalized contrastive loss to preserve both instance-level and state-level similarities across multiple temporal resolutions. To further capture temporal dynamics, we design a next-transition prediction pretext task that encourages representations to encode predictive information about future state evolution. We evaluate PLanTS across a wide range of downstream tasks—including classification, forecasting, trajectory tracking, and anomaly detection. PLanTS consistently improves the representation quality over existing SSL methods and demonstrates superior computational efficiency compared to baseline methods.

## 1 Introduction

Multivariate time series (MTS) data are now prevalent across a wide range of domains, including healthcare, climate science, and industrial monitoring (Zhang et al., 2018; Nguyen et al., 2017; Cook et al., 2019). However, MTS data is inherently high-dimensional, often non-stationary, and typically exhibit limited labeled instances, which presents significant challenges for supervised learning approaches (Montgomery et al., 2015; Cheng et al., 2015; Liu et al., 2022). In different application settings, tasks such as classification (Ismail Fawaz et al., 2019), forecasting (Lim & Zohren, 2021) and anomaly detection (Zamanzadeh Darban et al., 2024) often require extracting distinct and task-specific information from the temporal signals. Training task-specific model for each objective is not only computationally expensive but also lacks knowledge sharing across tasks.

To overcome these limitations, self-supervised learning (SSL) has emerged as a promising paradigm for learning general-purpose representations from unlabeled MTS data (Zhang et al., 2024; Trirat et al., 2024). Recent SSL methods typically rely on either handcrafted augmentations (Zheng et al., 2024) or context-based modeling (Yue et al., 2022; Lee et al., 2024) to construct positive and negative pairs for contrastive learning. These pairs are designed to encourage the model to learn representations that are invariant to noise and transformation, while preserving semantic similarity.

However, the effectiveness of the representations depends on the alignment between semantic similarity and the pairwise relationships constructed by the SSL methods (Wang et al., 2022; Demirel & Holz, 2024). Naive pairing strategies overlook the periodic structures inherent in real-world MTS data (Nagendra et al., 2011; Rhif et al., 2019), resulting in false positive and negative pairs that undermine the contrastive objective and diminish downstream performance. Furthermore, existing SSL methods generate instance-wise or timestamp-wise contrastive labels (Yue et al., 2022; Fraikin et al., 2024; Lee et al., 2024) that ignore the latent states and their temporal transitions. This is a critical limitation, as real-world MTS data involve non-stationary latent states whose dynamics

affect the observed signals over time (Tonekaboni et al., 2021). For example, in Human Activity Recognition (HAR) tasks using wearable sensors, the motion states (e.g., walking, sitting, running) in each individual are irregular and with variable durations (Figure 1). Similarly, identifying patients' disease progression states using MTS clinical record data is critical for disease management and decision making (Schulam et al., 2015; Suresh et al., 2018). In such cases, learning representations that not only discriminate between latent states but also capture the transitions between states are essential for accurately tracking, forecasting, and utilizing MTS data.

> Real-world MTS exhibit quasi-periodicity, nonstationarity, and multi-scale temporal dependencies. Local segments capture fine-grained fluctuations, while longer windows reveal latent regime transitions. Conventional point-level(Yue et al., 2022; Lee et al., 2024) or fixed-window(Tonekaboni et al., 2021) contrastive learning fails to respect these structures, causing misaligned similarity assignments and unstable representations under distributional shifts. Periodicity-aware multi-granularity modeling allows the SSL framework to (1) align patching with intrinsic rhythmicity in the data, (2) capture latent-state structure at multiple temporal resolutions, and (3) encode both short-term dynamics and long-term transitions.

To address the above challenges, we propose PLanTS, a Periodicity-aware Latent-state representation learning framework for robust and generalizable representation of complex, non-stationary MTS data. PLanTS introduces a multi-granularity generalized contrastive loss guided by varied periodic structures inferred from the input, based on the intuition that dominant periodic patterns often correspond to latent state transitions. Unlike conventional approaches that treat states as binary positive/negative pairs, PLanTS evaluates the similarity among latent states. In addition, PLanTS incorporates a pretext task to ensure that the learned embeddings encode predictive information of future state transitions, thereby explicitly modeling temporal dependencies across latent states.

We conduct a series of experiments across a wide range of downstream tasks, including multi-class and multi-label classification, forecasting, trajectory tracking, and anomaly detection. The benchmarking is conducted on five public MTS datasets, spanning healthcare, human activity recognition, energy systems, and web traffic domains. Our results demonstrate that PLanTS consistently improves the representation quality over existing SSL methods and achieves the best performance across diverse tasks compared to 13 baseline methods. Code is available at: `https://anonymous.4open.science/r/ICLR_2026_PLanTS-03DF/README.md`.

The key contributions of this work include:

- We propose PLanTS, a periodicity-aware, multi-granularity self-supervised learning framework for representing non-stationary multivariate time series. The embedding learned by PLanTS can be effectively applied in downstream MTS analysis tasks.

- In PLanTS, we introduce a generalized contrastive loss to effectively capture the periodic similarity for latent state representation; we also design a next transition prediction pretext task to model the temporal transition of latent states.

- PLanTS outperforms SOTA methods across four downstream tasks. We also demonstrated that the embedding learned by PLanTS more accurately captured the latent states and their transitions than baseline methods.

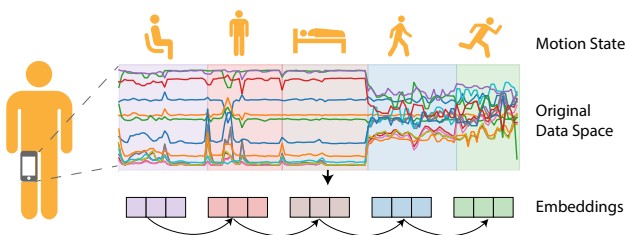

Figure 1: Human activity recognition tasks using wearable sensors. Background colors in the original data space indicate ground-truth motion states. PLanTS is designed to distinguish unknown motion states and model the dynamic transitions between them.

## 2 RELATED WORK

**Self-supervised learning.** Self-supervised learning has emerged as a powerful paradigm for extracting informative representations from unlabeled data by formulating pretext tasks that transform unsupervised objectives into supervised learning problems(Liu et al., 2021). In natural language processing, common pretext tasks include next-token prediction and masked-token prediction(Devlin et al., 2019; Rethmeier & Augenstein, 2023), while in computer vision, tasks such as solving jigsaw puzzles(Noroozi & Favaro, 2016), predicting image rotations(Gidaris et al., 2018) and clustering augmented views(Caron et al., 2018) have been widely adopted. More recently, contrastive learning-exemplified by frameworks such as SimCLR(Chen et al., 2020) and MoCo(He et al., 2020), has gained significant attention by constructing multiple views of the same instance and encouraging alignment of positive pairs while pushing apart negative pairs based on InfoNCE loss(Oord et al., 2018). However, many SSL methods developed for vision and language domains rely on domain-specific inductive biases, which are not directly applicable to time series data, where periodic structures and temporal continuity are critical.

**Contrastive learning for time series data.** Recent studies have demonstrated the effectiveness of contrastive learning (CL) in time series representation learning. T-loss (Franceschi et al., 2019) introduces a triplet-loss-based approach that employs time-based negative sampling for multivariate time series. TSTCC (Eldele et al., 2021) proposes a temporal and contextual contrasting framework that generates two related views via weak and strong augmentations. TF-C (Zhang et al., 2022) incorporates a time-frequency consistency mechanism to jointly learn time-domain and frequency-domain representations. While these methods focus primarily on instance-level contrast, they often struggle with temporally-sensitive downstream tasks such as forecasting. To address this, TS2Vec (Yue et al., 2022) introduces a hierarchical contrastive strategy that combines instance-wise and temporal-wise losses. T-Rep (Fraikin et al., 2024) further enhances temporal modeling by leveraging time-aware embeddings in the pretext task. SoftCLT (Lee et al., 2024) replaces the traditional hard contrastive objective with a soft contrastive loss. However, most existing methods neglect the inherent periodic structures present in real-world MTS. Moreover, approaches such as SoftCLT require a precomputed pairwise distance matrix, which becomes computationally prohibitive for long-term MTS data.

**Latent state representation in time series.** Latent states, such as motion states in human activity recognition (HAR) or clinical states in healthcare, play a crucial role in characterizing the dynamics of time series data. Learning how these states evolve over time is essential for capturing long-term trajectories and predicting future trends. To model such latent states, TNC (Tonekaboni et al., 2021) introduces the notion of temporal neighborhoods, treating temporally adjacent windows as positive pairs and distant windows as negative pairs. Time2State (Wang et al., 2023) proposes an unsupervised framework that applies a sliding window mechanism to extract distinguishable representations. However, existing methods focus primarily on identifying latent states in isolation and neglect the similarity and transitions between them. As a result, these approaches often yield coarse-grained representations that perform well for classification but generalize poorly to other downstream tasks.

## 3 METHODS

The overall framework of PLanTS is illustrated in Figure 2b. PLanTS is designed to learn a representation by modeling two components: the intrinsic variation within latent states and the transitions between states. Specifically, PLanTS consists of three main components: (1) a periodicity-aware multi-granularity patching module, which decomposes MTS data into structured patches aligned with dominant periodic patterns; (2) two dedicated encoders, namely the Latent State Encoder (LSE) and the Dynamic Transition Encoder (DTE), that complementarily capture the representations of within-state variations and state-to-state transitions; and (3) a fusion module, which integrates the latent state and transition embeddings for downstream tasks.

### 3.1 NOTATIONS AND PROBLEM DEFINITION

Consider a multivariate time series input $X = \{\mathbf{x}_1, \mathbf{x}_2, ..., \mathbf{x}_N\} \in \mathbb{R}^{N \times L \times C}$, where $N$, $L$ and $C$ denote the total number of samples, timestamps and channels respectively. The objective is to learn a non-linear embedding function $\mathcal{F}_\theta : \mathbb{R}^{L \times C} \to \mathbb{R}^{L \times D}$ to project each input sample $\mathbf{x}_i$ into a latent representation $\mathbf{z}_i \in \mathbb{R}^{L \times D}$, where $D$ is the embedding dimension. In PLanTS, $\mathcal{F}_\theta$ is composed of

two sub-modules: the Latent State Encoder $\mathcal{F}_L : \mathbb{R}^{L \times C} \to \mathbb{R}^{L \times D_l}$, which captures latent states and is learned via a multi-granularity generalized contrastive loss; and the Dynamic Transition Encoder $\mathcal{F}_T : \mathbb{R}^{L \times C} \to \mathbb{R}^{L \times D_t}$, which models temporal transitions between latent states using a novel self-supervised pretext task. The final representation is a concatenation of $\mathcal{F}_L(\mathbf{x}_i)$ and $\mathcal{F}_T(\mathbf{x}_i)$, namely, $\mathbf{z}_i = \left[ \mathcal{F}_L(\mathbf{x}_i) \,\|\, \mathcal{F}_T(\mathbf{x}_i) \right] \in \mathbb{R}^{L \times D}$, with $D = D_l + D_t$.

### 3.2 PERIODICITY-AWARE MULTI-GRANULARITY PATCHING MECHANISM

Real-world MTS data often comprises multiple latent states, making it challenging to capture transitions between intertwined latent states. Figure 2a compares contrastive learning paradigms from prior self-supervised learning methods with our proposed PLanTS framework. TNC (Tonekaboni et al., 2021)formulates a fixed window-based contrastive task, defining temporally neighboring windows as positives and distant ones as negatives, while TS2Vec (Yue et al., 2022) adopts a time-point–based contrastive formulation, encouraging contextual consistency at each timestamp. However, in practice, latent states can occur at diverse time scales. The reliance on a fixed window size or a time-point fails to capture the variability, substantially limiting applicability to real-world MTS data. To overcome this limitation, we introduce a periodicity-aware, multi-granularity patching approach that adaptively selects window sizes based on dominant periodic structures inferred from the input time series.

Real-world MTS data often comprises multiple latent states, making it challenging to capture transitions between intertwined latent states. Figure 2a compares contrastive learning paradigms from prior self-supervised learning methods with our proposed PLanTS framework. TNC (Tonekaboni et al., 2021)formulates a fixed window-based contrastive task, defining temporally neighboring windows as positives and distant ones as negatives, while TS2Vec (Yue et al., 2022) adopts a time-point–based contrastive formulation, encouraging contextual consistency at each timestamp. However, in practice, latent states can occur at diverse time scales. The reliance on a fixed window size or a time-point fails to capture the variability, substantially limiting applicability to real-world MTS data. To overcome this limitation, we introduce a periodicity-aware, multi-granularity patching approach that adaptively selects window sizes based on dominant periodic structures inferred from the input time series.

Inspired by Wu et al. (2022), we employ the Fast Fourier Transform (FFT) to identify prominent periodic patterns and determine appropriate time scales for patching. Basically, for each input $X$, we start by computing the channel-averaged amplitude spectrum:

$$F = \text{Avg}(\text{Amp}(\text{FFT}(X))), \quad f_1, \ldots, f_K = \arg \max_{f_* \in [1, \frac{L}{3}]}^{\text{Top-}K} (F), \quad w_j = \lceil \frac{L}{f_j} \rceil \qquad (1)$$

Here, $\text{FFT}(\cdot)$ denotes the Fourier transform applied along the temporal axis and $\text{Amp}(\cdot)$ computes the corresponding amplitude spectrum. To reduce the effect of high-frequency noise, we restrict attention to the lower-frequency index set $f_* = \{1, \ldots, \lfloor L/3 \rfloor\}$. We then order the amplitudes in descending magnitude, and denote $f_j$ as the frequency index attaining the $j$-th largest amplitudes. Each selected frequency $f_j$ is associated with a period length $w_j = \lceil \frac{L}{f_j} \rceil$, $j = 1, \ldots, K$, which is subsequently used as the window size in the dynamic-granularity patching module.

Given an input multivariate time series sample $\mathbf{x}_i \in \mathbb{R}^{L \times C}$ and the set of computed window sizes $\{w_1, \ldots w_K\}$, we treat each window size as a granularity and partition the input into non-overlapping patches of length $w_k$. Specifically, for granularity $w_k$, the input is divided into $M_k = \lceil \frac{L}{w_k} \rceil$ patches, denoted as $X_i^{(k)} = \{\mathbf{x}_{i,1}^k, \ldots, \mathbf{x}_{i,M_k}^k\}$, with $\mathbf{x}_{i,m}^k \in \mathbb{R}^{w_k \times C}, m = 1, \ldots M_k$, denoting the $m$-th patch at granularity $k$. Zero-padding is applied to ensure divisibility if necessary. The data patch $X_i^{patch} = \{X_i^{(1)}, \ldots, X_i^{(K)}\}$ are then fed into LSE and DTE to extract latent state representations and dynamic transition representations.

In this setting, each granularity reflects a distinct temporal resolution. PLanTS encodes these patches independently through the latent-state and dynamic-transition encoders, compute a contrastive loss for each of them, then integrate the information by taking the average of all these losses. This hierarchical structure enables the model to capture both within-state variations and between-state transitions consistently across scales.

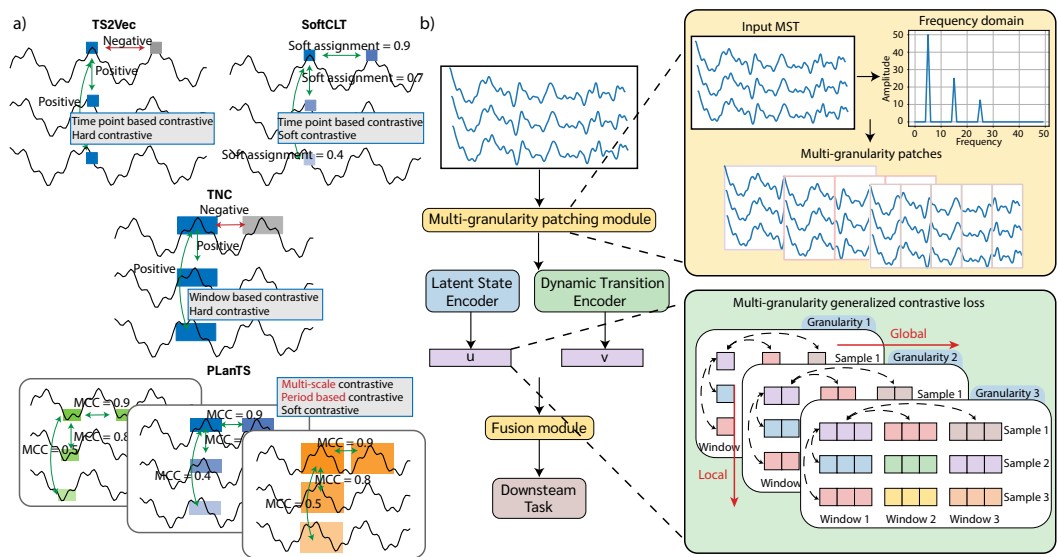

Figure 2: **Overview of the PLanTS framework.** a) Comparison with existing contrastive mechanisms for MTS. TS2Vec and SoftCLT utilize point-based contrastive learning, forming positive and negative pairs via contextual or soft assignment strategies. TNC applies a hard contrastive mechanism over fixed-size windows. In contrast, PLanTS incorporates periodic structure and introduces a multi-granularity, period-aware generalized contrastive learning framework that operates on dynamic latent states. b) Overall PLanTS framework.

### 3.3 LATENT STATES REPRESENTATION

To effectively capture latent states from multivariate time series, it is crucial to model the semantic similarity among different states. Conventional contrastive learning falls short because it reduces these relationships to binary labels, whereas in practical MTS data, they are continuous and hierarchical. We address this gap with a multi-granularity generalized contrastive loss that models both instance-level and state-level similarities across multiple temporal resolutions.

**Periodic Feature Similarity.** To capture the similarities between time series segments, SoftCLT Lee et al. (2024) relies on a precomputed dynamic time warping (DTW) distance matrix, which is computationally prohibitive for long-term multivariate time series data. Inspired by Yang et al. (2023), we compute Maximum Cross-Correlation (MXCorr) between time series windows in the input space.

Unlike DTW, which explicitly aligns sequences through dynamic programming, MXCorr measures the phase-invariant similarity between windows by finding the maximal normalized cross-correlation across possible temporal shifts. This property is particularly beneficial for quasi-periodic signals (e.g., ECG, sensor motion), where latent states may exhibit small phase shifts. By leveraging similarities captured through MXCorr, PLanTS effectively preserves latent state structures directly in the raw data space, providing informative self-supervision for robust latent-state representation learning.

Let $\mathbf{x}, \mathbf{y} \in \mathbb{R}^{w \times C}$ be two MTS patches sliced using the same window size $w$ with $C$ channels. The MXCorr between $\mathbf{x}$ and $\mathbf{y}$ is defined as:

$$\text{MXCorr}(\mathbf{x}, \mathbf{y}) = \frac{1}{C} \sum_{c=1}^{C} \max_{\tau \in [0, w-1]} \text{CC}(\mathbf{x}^{(c)}, \mathbf{y}^{(c)}; \tau) \tag{2}$$

, where $\text{CC}(\mathbf{x}^{(c)}, \mathbf{y}^{(c)}; \tau)$ represents the normalized cross-correlation between $\mathbf{x}$ and $\mathbf{y}$ shifted by time lag $\tau$ at the $c$-th channel (see details in Appendix D). We implemented an efficient batch computation of MXCorr, and assessed its efficiency by comparing with SoftCLT, as detailed in Appendix E.

**Local instance-wise contrastive learning.** In real-world MTS, latent states encode identity-specific characteristics—i.e., even when two time series samples are in the same latent state, their representations should remain distinguishable due to individual variations. To capture this individual difference, PLanTS incorporates a local instance-wise contrastive loss that models variations among samples within the same time window.

Denote $\mathbf{u}_i \in \mathbb{R}^{M_k \times w_k \times D_l}$ as the latent state embeddings of the $i$-th time series sample $X_i^{(k)} \in \mathbb{R}^{M_k \times w_k \times C}$ patched at $k$-th granularity. PlanTS treats all other samples in the batch as negative views weighted by input-space feature similarity. Specifically, for the $i$-th time series sample in the $m$-th window, the local instance-wise contrastive loss is formulated as:

$$l_{\text{local}}^{i,m} = - \sum_{j=1,j\neq i}^{B} \frac{\exp(\mathbf{s}_{ij}^m)}{\sum_{j'=1,j'\neq i}^{B} \exp(\mathbf{s}_{ij'}^m)} \log \frac{\exp(\langle \mathbf{u}_i^m \cdot \mathbf{u}_j^m \rangle)}{\sum_{j'=1,j'\neq i}^{B} \exp(\langle \mathbf{u}_i^m \cdot \mathbf{u}_{j'}^m \rangle)} \tag{3}$$

Here, $B$ is batch size, $\mathbf{s}_{ij}^m = \text{MXCorr}(\mathbf{x}_{i,m}^k, \mathbf{x}_{j,m}^k)$ denotes the input-space similarity between samples $i$ and $j$ at the $m$-th window, $i = 1, \ldots, N; j = 1, \ldots, N; m = 1, \ldots, M_k$, and $\langle \cdot, \cdot \rangle$ denotes inner product. This formulation extends the InfoNCE loss Oord et al. (2018), which assigns equal weight to all negative pairs. In contrast, our method employs a soft weighting scheme based on input-space similarity $\mathbf{s}_{ij}$, encouraging alignment between representations based on the similarity in the original space. We prove in Appendix D that this weighted contrastive loss is equivalent to minimizing the KL divergence between the predicted softmax distribution and the similarity-based target distribution.

**Global state-wise contrastive learning.** Beyond capturing local variations in short temporal windows, robust representation learning for MTS data requires modeling long-term evolutions. PLanTS addresses this issue by introducing a global state-wise contrastive loss that explicitly captures continuous relationships among latent states along the temporal axis.

Denote $\mathbf{u}_i^m \in \mathbb{R}^{w_k \times D_l}$ as the latent state representation of the $m$-th window from the $i$-th time series sample at granularity $k$. Similar to the local instance-wise contrastive loss, PLanTS compares this window against all other windows from the same sample. The global contrastive loss for the $m$-th window is defined as:

$$l_{\text{global}}^{i,m} = - \sum_{n=1,n\neq m}^{M_k} \frac{\exp(\mathbf{a}_{mn}^i)}{\sum_{n'=1,n'\neq m}^{M_k} \exp(\mathbf{a}_{mn'}^i)} \log \frac{\exp(\langle \mathbf{u}_i^m \cdot \mathbf{u}_i^n \rangle)}{\sum_{n'=1,n'\neq m}^{M_k} \exp(\langle \mathbf{u}_i^m \cdot \mathbf{u}_i^{n'} \rangle)} \tag{4}$$

, where $\mathbf{a}_{mn}^i = \text{MXCorr}(\mathbf{x}_{i,m}^k, \mathbf{x}_{i,n}^k)$ denotes the similarity between the $m$-th and $n$-th windows of sample $i$ at granularity $k$.

**Overall contrastive learning loss.** The overall contrastive loss for the $k$-th granularity is the joint of the local and global contrastive losses:

$$L_l^{(k)} = \frac{1}{N \cdot M_k} \sum_{i=1}^{N} \sum_{m=1}^{M_k} \left( \alpha \cdot l_{\text{local}}^{i,m} + (1-\alpha) \cdot l_{\text{global}}^{i,m} \right) \tag{5}$$

, where $\alpha$ is a hyperparameter controlling the contribution of each loss.

### 3.4 DYNAMIC TRANSITION REPRESENTATIONS

Beyond learning representations that distinguish among latent states, it is essential to model the state transitions to effectively track and forecast latent state trajectories in MTS data. To this end, we introduce a next-transition prediction pretext task to encourage the model to encode predictive information about latent states transition.

**Next transition prediction.** In real-world MTS data, temporal variations often manifest as shifts between latent states. For example, fluctuations in a patient's vital signs may reflect disease progression, which can be viewed as state transitions. To model such dynamics, we propose a next-transition prediction task that aims at forecasting future transitions conditioned on both the current latent state and its dynamic transition representation.

Given a time series sample $X_i^{(k)} \in \mathbb{R}^{M_k \times w_k \times C}$ at $k$-th granularity, The Dynamic Transition Encoder $\mathcal{F}_T$ outputs dynamic transition embedding: $\mathbf{v}_i = \mathcal{F}_T(x_i^{(k)}) \in \mathbb{R}^{M_k \times w_k \times D_t}$. At each window $m$, we concatenate the latent state representation $\mathbf{u}_i^m$ and dynamic transition representation $\mathbf{v}_i^m$, and feed the result into a prediction head $G : \mathbb{R}^{D_l + D_t} \to \mathbb{R}^{D_t}$, which is implemented as a two-layer MLP with ReLU activations. The objective is to minimize the mean squared error (MSE) between the predicted next transition and the ground-truth transition at window $m + 1$:

$$L_t^{(k)} = \frac{1}{N \cdot (M_k - 1)} \sum_{i=1}^{N} \sum_{m=1}^{M_k-1} \left| G\left(\text{concat}(\mathbf{u}_i^m, \mathbf{v}_i^m)\right) - \mathbf{v}_i^{m+1} \right|^2 \tag{6}$$

This loss term encourages the model to encode the predictable transitions between latent states, enabling temporal-aware representational learning of MTS data.

**Final Objective.** The overall loss function of PLanTS combines both the loss terms of latent state representation and dynamic transition representation across all granularities:

$$L = \frac{1}{K} \sum_{k=1}^{K} \left( \lambda L_l^{(k)} + (1 - \lambda) L_t^{(k)} \right) \tag{7}$$

, where $\lambda$ is a hyperparameter controlling the contribution of each loss.

## 4 EXPERIMENTS

To evaluate the performance of PLanTS, We conduct a series of experiments across diverse downstream tasks for MTS: (1) multi-class classification, (2) multi-label classification, (3) forecasting, and (4) anomaly detection (see details in Appendix F). In addition, we perform ablation studies to assess the contribution of each core component in PLanTS. Finally, we analyze the temporal trajectories of the learned representations to better understand how latent state transitions are captured and encoded in the representation space. Detailed experimental setups, additional results, and further analysis are provided in the Appendix B.

### 4.1 MULTI-CLASS CLASSIFICATION

We evaluate the instance-level representations learned by PLanTS on 30 benchmark datasets from the UEA multivariate time series classification archive Bagnall et al. (2018), covering diverse domains such as healthcare, sensor systems, speech, and human activity recognition. We compare PLanTS with 9 SOTA self-supervised learning baselines: DTW Chen et al. (2013), TST Zerveas et al. (2021), TS-TCC Eldele et al. (2021), T-Loss Franceschi et al. (2019), TNC Tonekaboni et al. (2021), TS2Vec Yue et al. (2022), CSLLiang et al. (2023), T-Rep Fraikin et al. (2024), and SoftCLT Lee et al. (2024). Following the evaluation protocol of TS2Vec, we train an SVM classifier with an RBF kernel on top of the learned representations to perform classification.

The evaluation results are summarized in Table 1 and the full results are provided in Appendix G. We report the average rank (AR) of each algorithm across all datasets and the number of first-place finishes. For each pairwise comparison, we also compute the counts of datasets in which PLanTS wins, ties, or loses (W/T/L) against each counterpart. Statistical significance is assessed using the Wilcoxon signed-rank test, and the corresponding p-values (p-val) are reported. PLanTS achieves consistent and substantial improvements over all baselines, increasing average classification accuracy by 4.35% over TS2Vec and by 3.90% and 3.15% over T-Rep and CSL, respectively. PLanTS significantly outperforms all competing methods on the UEA datasets (p-value $< 0.05$ under most circumstances). It also achieves the highest number of first-place finishes, underscoring its strong performance on MTS classification.

### 4.2 MULTI-LABEL CLASSIFICATION

Unlike multi-class classification, multi-label classification does not assume class exclusivity, where multiple conditions can occur simultaneously. Thus, it provides a more realistic and stringent evalu-

Table 1: Summary of classification results on the 30 UEA MTS archive.

| Method | Avg. Acc. | Avg. Rank | Ranks 1st | Avg. Diff. (%) | W/T/L | Wilcoxon P-value |
|--------|-----------|-----------|-----------|----------------|-------|------------------|
| DTW | 0.650 | 5.862 | 1 | 9.214 | 21/1/7 | 0.001 |
| TST | 0.617 | 6.900 | 2 | 10.863 | 24/2/4 | 0 |
| TS-TCC | 0.668 | 5.633 | 4 | 6.463 | 20/2/8 | 0.001 |
| T-Loss | 0.658 | 4.833 | 5 | 7.670 | 17/4/9 | 0.009 |
| TNC | 0.670 | 5.767 | 2 | 5.640 | 23/3/4 | 0 |
| TS2Vec | 0.690 | 5.100 | 3 | 5.063 | 21/0/9 | 0.003 |
| T-Rep | 0.693 | 4.667 | 4 | 5.430 | 18/5/7 | 0.036 |
| SoftCLT | 0.709 | 4.481 | 5 | 4.544 | 16/1/10 | 0.033 |
| CSL | 0.698 | 4.867 | 7 | 4.544 | 18/0/12 | 0.089 |
| PLanTS | **0.720** | **3.333** | **8** | – | – | – |

Table 2: Performance comparison on PTB-XL multi-label classification tasks

| Task | Method | Accuracy | F1 Score | AUROC |
|------|--------|----------|----------|-------|
| Diagnostic | Ts2Vec | 0.447 | 0.594 | 0.825 |
| | T-rep | 0.440 | 0.558 | 0.836 |
| | SimCLR + DBPM | **0.458** | 0.583 | 0.806 |
| | PLanTS | **0.458** | **0.601** | **0.852** |
| Form | Ts2Vec | 0.366 | 0.509 | 0.768 |
| | T-rep | 0.311 | 0.482 | 0.744 |
| | SimCLR + DBPM | 0.349 | 0.480 | 0.752 |
| | PLanTS | **0.385** | **0.514** | **0.784** |
| Rhythm | Ts2Vec | 0.791 | 0.825 | 0.833 |
| | T-rep | **0.819** | **0.853** | 0.833 |
| | SimCLR + DBPM | 0.808 | 0.837 | 0.838 |
| | PLanTS | **0.819** | 0.852 | **0.863** |

ation by requiring models to capture overlapping patterns. We evaluate PLanTS on PTB-XLWagner et al. (2020), the largest publicly available clinical ECG waveform dataset, which includes three multi-label classification tasks: Diagnostic (44 classes), Form (19 classes), and Rhythm (12 classes).

We formulate the evaluation protocol by training a One-vs-Rest SVM classifier with an RBF kernel on top of the learned representations. We compare PLanTS against three SOTA self-supervised learning methods: Ts2Vec, T-Rep, and DBPMLan et al. (2024), a recently proposed SSL approach specifically designed for multi-label tasks. We employ four evaluation metrics: AUROC (macro-averaged), accuracy, F1 score (micro-averaged), and per-class AUROC. Results are reported in Table 2 and detailed in Appendix G.

PLanTS consistently achieves superior performance in AUROC, improving from 0.836 to 0.852 on the Diagnostic task, from 0.768 to 0.784 on the Form task, and from 0.838 to 0.863 on the Rhythm task. For both Diagnostic and Form classification, PLanTS surpasses all baselines across every metric, with accuracy gains from 0.447 to 0.458 and from 0.366 to 0.385, respectively. In the Rhythm task, PLanTS attains the highest AUROC while remaining competitive in accuracy and F1 score. Figure 3 illustrates the per-class AUROC for 10 diagnostic categories: whereas baseline methods suffer notable drops (e.g., DBPM on "AMI," T-Rep on "INJIL," TS2Vec on "LAO/LAE"), PLanTS achieves consistently high AUROC across all categories, underscoring its robustness and reliability in capturing fine-grained clinical semantics from multivariate ECG data.

Need to mention that: in the PTB-XL experiments, we set the hyperparameter $\lambda = 1$ (see Appendix B implementation details), disabling the next-transition prediction loss in the total objective. Because PTB-XL samples are short (10s) and do not exhibit meaningful latent transitions within each recording, making the dynamic transition pretext task less informative.

## 4.3 FORECASTING

We evaluate PLanTS on the MTS forecasting task using five benchmark datasets from the ETT suite—ETTh1, ETTh2, ETTm1, ETTm2 Zhou et al. (2021) and weather datasetMax Planck Institute for Biogeochemistry (2025). Following the standard evaluation protocol used by TS2Vec (Yue et al., 2022), we freeze the pretrained encoder and fit a ridge regression head on top of the learned

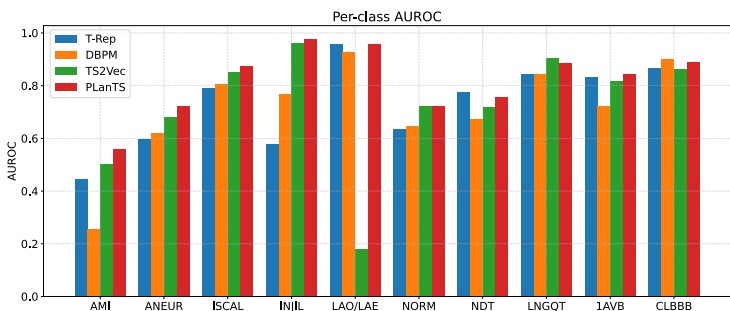

Figure 3: Per-class AUROC comparison on 10 selected diagnostic classes from PTB-XL.

Table 3: Forecasting performance on the ETT benchmark.

| | PLanTS | | SoftClt | | T-rep | | TS2Vec | | Informer | | TCN | |
|---|---|---|---|---|---|---|---|---|---|---|---|---|
| Dataset | MSE | MAE | MSE | MAE | MSE | MAE | MSE | MAE | MSE | MAE | MSE | MAE |
| ETTh1 | **0.708** | **0.621** | 0.836 | 0.670 | 0.763 | 0.645 | 0.803 | 0.665 | 0.907 | 0.739 | 1.021 | 0.816 |
| ETTh2 | 1.685 | 0.967 | **1.494** | **0.925** | 1.818 | 1.034 | 1.802 | 1.022 | 2.371 | 1.199 | 2.574 | 1.265 |
| ETTm1 | **0.531** | **0.507** | 0.628 | 0.547 | 0.584 | 0.529 | 0.631 | 0.565 | 0.749 | 0.640 | 0.818 | 0.849 |
| ETTm2 | 0.885 | 0.581 | **0.645** | **0.577** | 0.783 | 0.603 | 0.784 | 0.607 | 1.173 | 0.702 | 3.635 | 1.891 |
| Weather | **0.262** | **0.343** | 0.285 | 0.349 | 0.287 | 0.353 | 0.522 | 0.448 | 0.634 | 0.548 | 0.631 | 0.602 |
| Avg. Rank | **1.75** | **1.75** | 2.96 | 2.96 | 2.88 | 2.88 | 3.92 | 3.92 | 4.54 | 4.54 | 5.96 | 5.96 |
| Ranks 1st | **13** | **13** | 5 | 5 | 2 | 2 | 1 | 1 | 1 | 1 | 1 | 1 |

representations to forecast multiple future horizons and report the averaged forecasting performance across all horizons. PLanTS is compared with SOTA methods such as TNC, TS2Vec, T-Rep, Soft-CLT, and InformerZhou et al. (2021). The average forecasting performances for each horizon, average rank, and number of rank first over all datasets and prediction horizons are presented in Table 3 (full results are in Appendix G).

Overall, PLanTS achieves the best average performance, ranking first in 11 out of 16 settings (MSE) and 12 out of 16 settings (MAE). It consistently outperforms baseline methods on ETTh1 and ETTm1 evaluated by both MSE and MAE. On ETTh1 and ETTm1, PLanTS reduces the average MSE by 7.2% and 9.1%, and reduces MAE by 3.7% and 4.2%, respectively, compared to the best-performing baseline (T-Rep). PLanTS also achieves competitive results on ETTh2. Our results demonstrate the PLanTS's effectiveness in modeling fine-grained periodic and dynamic patterns for forecasting tasks. However, PLanTS does not perform as well on ETTm2 under MSE. One reason could be the higher level of noise and abrupt fluctuations in the ETTm2 data, which may decrease the quality of periodicity extraction and weaken the predictive strength of latent state transitions.

## 4.4 TRAJECTORY TRACKING

To investigate the latent space structure and validate that PLanTS captures irregular latent states, we evaluated it on the Human Activity Recognition (HAR) dataset from the UCI Machine Learning Repository (Anguita et al., 2013). UCI-HAR contains smartwatch-based recordings of 30 individuals performing six activities: walking, walking upstairs, walking downstairs, sitting, standing, and lying down. Activity switches provide ground-truth latent state transitions. Following (Tonekaboni et al., 2021), we constructed continuous trajectories by concatenating each individual's activity segments, enabling the analysis of state transitions in a realistic and temporally consistent manner.

To demonstrate that the embeddings learned by PLanTS capture latent state transition, we visualized the top three principal components of the learned embeddings and compared them with embeddings from TS2Vec and SoftCLT. As shown in Figure 4, the embeddings by PLanTS have sharper transitions and more distinct activity-specific patterns. In particular, PLanTS better separates similar motion states such as sitting and standing (marked in red and cyan in the time series sample trajectory), which cannot be identified by baseline methods or be directly seen in the original MTS signals.

Table 4: Ablation study of PLanTS in forecasting and classification benchmarks.

| Variant | Forecasting (↓ MSE) | | | | Classification (↑ Accuracy) | | | |
|---|---|---|---|---|---|---|---|---|
| | ETTh1 | ETTh2 | ETTm1 | ETTm2 | StandWalkJump | Heartbeat | RacketSports | Handwriting |
| PLanTS | 0.729 | 1.796 | 0.595 | 0.844 | **0.667** | **0.746** | **0.842** | **0.439** |
| w/o multi-granularity patching | **0.708** | **1.685** | **0.531** | 0.885 | 0.333 | 0.741 | 0.803 | 0.426 |
| w/o local contrastive | 0.795 | 1.916 | 0.571 | **0.826** | 0.200 | 0.746 | 0.796 | 0.165 |
| w/o global contrastive | 0.732 | 1.815 | 0.594 | 0.843 | 0.400 | 0.692 | 0.829 | 0.431 |
| w/o NTP | 0.735 | 1.918 | 0.571 | 0.849 | 0.333 | 0.737 | 0.829 | 0.291 |

The blue boxes in Figure 4 highlight two states that are clearly separated in the embedding learned by PLanTS, but remain indistinguishable in the representations of TS2Vec and SoftCLT. Additional results are provided in Appendix G. The results demonstrates PLanTS's ability to model latent state transitions—an essential property for post-hoc analysis and downstream applications in healthcare.

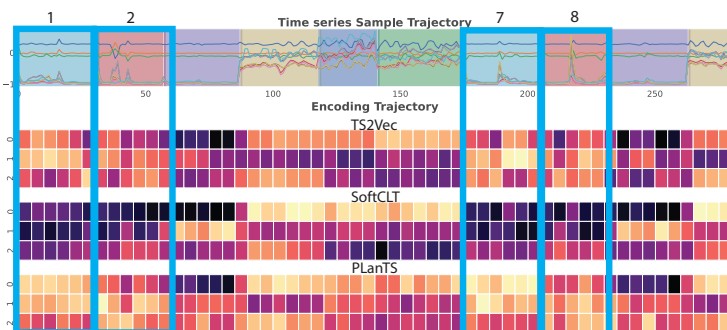

Figure 4: The top-3 PCs of a HAR signal trajectory encoded by TS2Vec, SoftCLT, and PlanTS. Only the embedding learned by PLanTS captured the transition between states 1-2 and states 7-8.

### 4.5 ABLATION STUDY

To assess the contribution of each component in PLanTS, we conducted comprehensive ablation studies on four forecasting datasets and four classification datasets. We compared the full version of PLanTS with the following variations: **w/o multi-granularity patching:** removes the periodicity-aware multi-granularity patching mechanism and segments inputs into non-overlapping patches using a fixed window size of 50. **w/o local contrastive:** disables the local instance-wise contrastive loss by setting $\alpha = 0$. **w/o global contrastive:** disables the global state-wise contrastive loss by setting $\alpha = 1$. **w/o NTP:** removes the next transition prediction pretext task by setting $\lambda = 1$.

Table 4 details the ablation results. The multi-granularity patching mechanism is critical for classification, with its removal causing large accuracy drops (e.g., –50.07% on StandWalkJump), while a single fixed-size strategy slightly benefits forecasting (MSE reductions of 2.88–10.76% on ETT datasets), likely due to their large periodicities. Contrastive losses and the next-transition prediction (NTP) objective are also essential: removing the local contrastive loss yields the steepest classification declines (–60.02% on StandWalkJump, –62.41% on Handwriting), eliminating the global loss reduces accuracy by 7.24% on Heartbeat, and discarding NTP lowers accuracy by 33.71% on Handwriting and increases MSE by 6.79% on ETTh2.

## 5 CONCLUSION

We propose PLanTS, a self-supervised framework for learning latent state representations in non-stationary MTS data. To capture irregular latent states, we introduce a periodicity-guided multi-granularity contrastive loss that preserves both instance-level and state-level similarities across multiple temporal resolutions. To further model state transitions, we design a next-transition prediction pretext task that encourages the representations to encode predictive transition dynamics. Extensive experiments across classification, forecasting, trajectory tracking, and anomaly detection demonstrate consistent performance improvements. PLanTS effectively encodes, tracks, and predicts latent states, making it broadly applicable to domains such as healthcare and human activity monitoring.

## REPRODUCIBILITY STATEMENT

The complete source code for PLanTS, can be seen in an anonymous link: `https://anonymous.4open.science/r/ICLR_2026_PLanTS-03DF/README.md` and will be made publicly available on GitHub upon publication. All datasets used in this work are publicly available, including 30 UEA, ETT (ETTh1, ETTh2, ETTm1, ETTm2), UCI-HAR, PTB-XL and Yahoo. We provide preprocessing scripts, configuration files, and documented hyperparameters (Appendix B) to facilitate exact replication.

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

Table 5: Dataset description.

| Dataset | Train | Val | Test | Channels | Length | Categories |
|---|---|---|---|---|---|---|
| PTB-XL Diagnostic | 13688 | 3422 | 4278 | 12 | 1000 | 44 |
| PTB-XL Form | 5745 | 1437 | 1796 | 12 | 1000 | 19 |
| PTB-XL Rhythm | 13459 | 3365 | 4206 | 12 | 1000 | 12 |
| HAR | 21 | – | 9 | 561 | 281,288 | 6 |

## A  DATASET DESCRIPTIONS

**Human Activity Recognition (HAR) dataset**

The UCI HAR dataset (Anguita et al., 2013)is a widely used benchmark for human activity recognition tasks. It consists of sensor data collected from 30 subjects aged 19–48 while performing six activities of daily living: walking, walking upstairs, walking downstairs, sitting, standing, and laying. Each subject wore a Samsung Galaxy S II smartphone on their waist, which recorded tri-axial linear acceleration and angular velocity at a sampling rate of 50 Hz. The raw signals were segmented into fixed-width windows of 2.56 seconds (128 time steps) with a 50% overlap. For each window, a set of 561 handcrafted time- and frequency-domain features was extracted. The dataset is split into training and test sets based on subject IDs. In our trajectory tracking experiment, we construct continuous activity trajectories for each subject by concatenating their activity sequences based on subject identifiers. Details are shown in Table 5.

**PTB-XL ECG Database**

PTB-XL is a large-scale, publicly available electrocardiogram (ECG) dataset (Wagner et al., 2020) published by the PhysioNet initiative. It contains 21,837 clinical 12-lead ECG records, each lasting 10 seconds and sampled at 500 Hz, from 18,885 unique patients. The dataset includes diagnostic annotations covering multiple labeling dimensions such as diagnostic, form, and rhythm classes, enabling both single- and multi-label classification tasks. Altogether, there are 71 distinct statements, comprised of 44 diagnostic, 12 rhythm, and 19 form statements, with 4 of these also serving as diagnostic ECG statements. Based on the ECG annotation method, there are three multi-label classification tasks: Diagnostic Classification (44 classes), Form Classification (19 classes), and Rhythm Classification (12 classes). We use data spliting rate 0.6,0.2,0.2 to split training, testing and validation sets and follow the data pre-processing steps from Lan et al. (2024). Table 5 provides a summarization of PTB-XL dataset.

**Yahoo dataset**

Yahoo datasetRen et al. (2019) is a widely used benchmark for time-series anomaly detection, containing 367 synthetic and real-valued univariate time series grouped into four subsets (A1–A4), each labeled with point-wise anomalies. For fair comparison, we follow the same evaluation strategy as Yue et al. (2022). The anomalies detected within a certain delay (7 steps for minutely data and 3 steps for hourly data) are considered correct. Additionally, during preprocessing, the raw time series is differenced $d$ times to mitigate non-stationary drift, where $d$ is the number of unit roots estimated using the Augmented Dickey-Fuller (ADF) test.

## B  IMPLEMENTATION DETAILS

The models are implemented in Python 3.12.11, using PyTorch 2.3.0 for deep learning and scikit-learn for SVMs, linear regressions, and data pre-processing. We employ the Adam optimizer in all experiments. Training is conducted on AWS g5 xlarge and g5 2xlarge instances, each equipped with NVIDIA A10G GPUs, using CUDA 11.6.

**Encoder architecture.** The PlanTS encoder consists of two parallel components: a Latent State Encoder (LSE) and a Dynamic Transition Encoder (DTE). Both modules follow a deep dilated convolutional architecture. Each branch first projects the input sequence through a fully connected layer (64 dimensions), followed by a stack of 10 residual convolutional blocks with exponentially increasing dilation factors (from $2^0$ to $2^9$), GELU activations, and skip connections. LSE and DTE outputs representations of dimension 128; both are regularized with dropout ($p = 0.1$).

Table 6: Hyperparameter settings for various tasks.

| Hyperparameter | Classification | Trajectory tracking | Anomaly detection | Multi-label classification | Forecasting |
|---|---|---|---|---|---|
| $(\alpha, \lambda)$ | (0.5,0.5),(0.9,1) | (0.5,0.5) | | (0.9,1) | (0.5,0.5) |
| $K$ | | 3 | | window size=[20,30] | window size=50 |
| $lr$ | 0.0001-0.001 | | 0.001 | | |
| $bs$ | | | 128 | | |

**Hyperparameters.** The hyperparameter configurations used in our experiments are summarized in Table 6. There are five hyperparameters used in PLanTS: $\alpha$, $\lambda$, $K$, window size, learning rate ($lr$), and batch size ($bs$). Here, $\alpha$ and $\lambda$ control the relative contributions of the local contrastive, global contrastive, and next-transition prediction losses; we report them as pairs. $K$ denotes the number of dominant periodicities used in the period-aware multi-granularity patching strategy. When this mechanism is not applied, we instead report the fixed window size used. $lr$ represents learning rate and $bs$ denotes batch size. For $(\alpha, \lambda)$, we select from $\{(0.5, 0.5), (0.9, 1)\}$ depending on the task. We apply the period-aware multi-granularity patching mechanism in the *Classification* and *Trajectory Tracking* tasks, setting $K = 3$. For *Multi-label Classification* and *Forecasting*, we replace $K$ with fixed window sizes: [20, 30] for multi-label classification and 50 for forecasting. The learning rate is fixed at 0.001 for all tasks except *Classification*, where we sweep from 0.0001 to 0.001 to ensure convergence across all 30 UEA datasets. The batch size is set to 128 for all experiments.

## C  HYPER-PARAMETER SENSITIVITY

We evaluate the sensitivity of PLanTS to the hyperparameters $\alpha$ and $\lambda$ (introduced in Equations 5 and 7), which control the relative weights of the loss terms. Figures 5 and 6 report the relative percentage change in MSE and MAE with respect to the best results across four forecasting datasets. Overall, PLanTS exhibits stable performance under a wide range of hyperparameter values, demonstrating the robustness of the framework. We also observe that $\lambda$, which balances the latent state representation loss against the dynamic transition loss, has a stronger influence on performance—particularly on ETTh2—suggesting that accurately modeling transition dynamics is critical for datasets with more complex temporal dependencies.

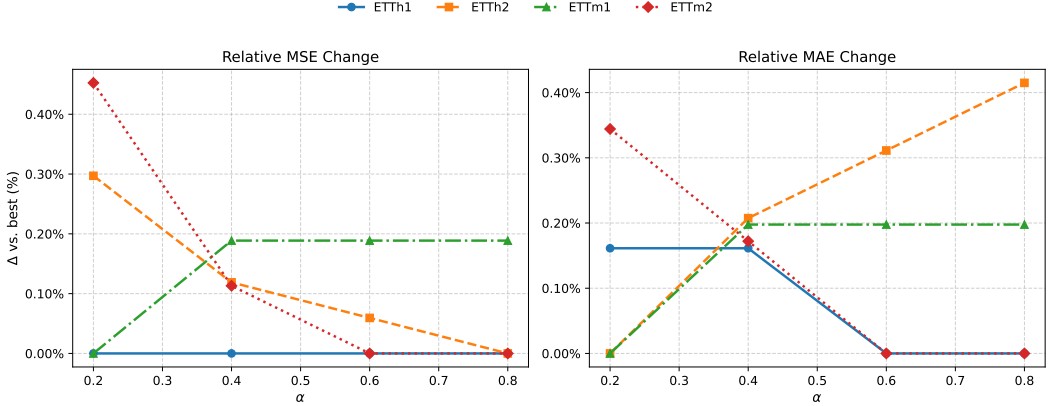

Figure 5: Sensitivity analysis of hyper-parameters $\alpha$ in forecasting task.

## D  DERIVATION OF WEIGHTED CONTRASTIVE LOSS

The cross-correlation for each channel $c$ is computed via FFT as:

$$\mathrm{CC}_c(x^{(c)}, y^{(c)}; \tau) = \mathcal{F}^{-1}\left(\frac{\mathcal{F}(x^{(c)} - \bar{x}^{(c)}) \cdot \overline{\mathcal{F}(y^{(c)} - \bar{y}^{(c)})}}{\sigma_x^{(c)} \cdot \sigma_y^{(c)} + \varepsilon}\right)_\tau \quad (8)$$

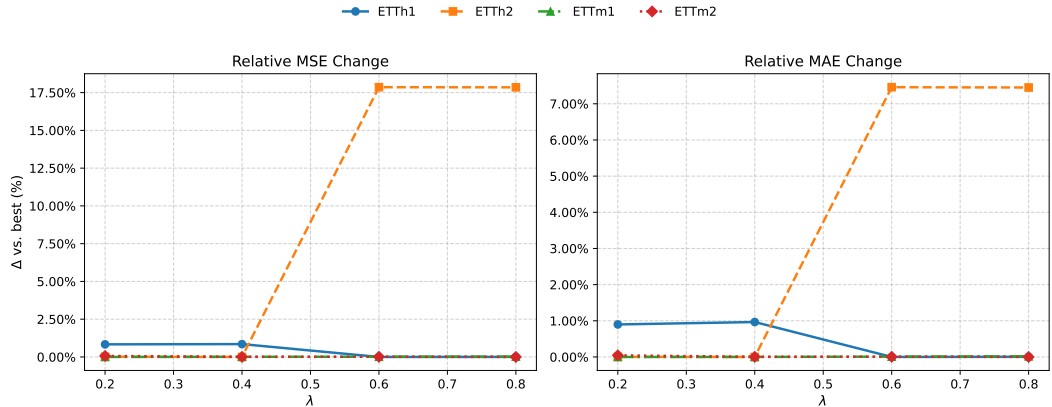

Figure 6: Sensitivity analysis of hyper-parameters $\lambda$ in forecasting task.

In this section, we aim at demonstrating that minimizing our weighted contrastive loss is equivalent to minimizing the KL divergence between the predicted softmax distribution and the similarity-based target distribution. We define the predicted softmax distribution as $Q$, and define the input-space similarity distribution measured by Maximum Cross-Correlation as $P$, where:

$$\mathrm{q}(i,j) = \mathrm{Q}_{ij} = \frac{\exp(u_i^m \cdot u_j^m)}{\sum_{j'=1, j'\neq i}^{B} \exp(u_i^m \cdot u_{j'}^m)} \tag{9}$$

$$\mathrm{p}(i,j) = \mathrm{P}_{ij} = \frac{\exp(s_{ij}^m)}{\sum_{j'=1, j'\neq i}^{B} \exp(s_{ij'}^m)} \tag{10}$$

Then the local instance-wise contrastive loss can be formulated as:

$$l_{\mathrm{local}}^{i,m} = -\sum_{j=1, j\neq i}^{B} \mathrm{p}(i,j) \log \mathrm{q}(i,j)$$
$$= \sum_{j=1, j\neq i}^{B} \left(\mathrm{p}(i,j)\log \mathrm{p}(i,j) - \mathrm{p}(i,j)\log \mathrm{q}(i,j)\right) - \sum_{j=1, j\neq i}^{B} \mathrm{p}(i,j)\log \mathrm{p}(i,j)$$
$$= \sum_{j=1, j\neq i}^{B} \mathrm{p}(i,j)\log \frac{\mathrm{p}(i,j)}{\mathrm{q}(i,j)} - \sum_{j=1, j\neq i}^{B} \mathrm{p}(i,j)\log \mathrm{p}(i,j)$$
$$= \mathrm{KL}(Q\|P) + \mathrm{constant}$$

## E  COMPUTATIONAL COMPARISON

To show the computation efficiency of PLanTS, we compare the running time of our method with one hard contrastive learning method Ts2Vec and one weighted contrastive learning method Spft-CLT. All experiments are conducted on simulated data under controlled settings. For fairness, we adopt the TS2Vec backbone architecture, set the batch size to 128 for all methods, and use a single-granularity strategy for PLanTS in this comparison. To better investgate the effect of sequence length $L$, number of samples $N$ and number of channels $C$ to running time, we keep two other variables fixed (e.g., 5,000 samples, 3 channels), vary only one variable (e.g. sequence length). Runtime is decomposed into *precomputation time*, *training time*, and *total runtime* (sum of both).

Tables 7 summarize the results. For all settings, TS2Vec achieves the lowest overall runtime due to its hard contrastive strategy. When comparing PLanTS to SoftCLT, we observe a consistent advantage in total runtime despite PLanTS operating fully end-to-end without any precomputation

Table 7: End-to-end runtime comparison on simulated data under varying sequence length $L$, number of samples $N$, and number of channels $C$. All times are in **seconds**. Precomp: precomputation time. Train: training time. Total: sum of both.

| Varied | Value | Method | Precomp | Train | Total |
|--------|-------|--------|---------|-------|-------|
| $L$ | 256 | TS2Vec | – | $26.34_{\pm 1.32}$ | $26.34_{\pm 1.32}$ |
| | | PLanTS | – | $110.93_{\pm 0.61}$ | $110.93_{\pm 0.61}$ |
| | | SoftCLT | $380.78_{\pm 2.52}$ | $335.17_{\pm 4.78}$ | $715.01_{\pm 8.88}$ |
| | 512 | TS2Vec | – | $57.55_{\pm 0.81}$ | $57.55_{\pm 0.81}$ |
| | | PLanTS | – | $166.26_{\pm 0.27}$ | $166.26_{\pm 0.27}$ |
| | | SoftCLT | $24.90_{\pm 0.09}$ | $447.17_{\pm 4.19}$ | $472.08_{\pm 4.11}$ |
| | 1024 | TS2Vec | – | $98.03_{\pm 3.02}$ | $98.03_{\pm 3.02}$ |
| | | PLanTS | – | $234.07_{\pm 0.37}$ | $234.07_{\pm 0.37}$ |
| | | SoftCLT | $25.99_{\pm 0.17}$ | $624.23_{\pm 2.40}$ | $650.23_{\pm 2.57}$ |
| $N$ | 100 | TS2Vec | – | $17.62_{\pm 2.80}$ | $17.62_{\pm 2.80}$ |
| | | PLanTS | – | $45.31_{\pm 0.85}$ | $45.31_{\pm 0.85}$ |
| | | SoftCLT | $1.21_{\pm 0.35}$ | $91.28_{\pm 7.57}$ | $92.48_{\pm 7.92}$ |
| | 500 | TS2Vec | – | $57.55_{\pm 0.81}$ | $57.55_{\pm 0.81}$ |
| | | PLanTS | – | $166.26_{\pm 0.27}$ | $166.26_{\pm 0.27}$ |
| | | SoftCLT | $24.90_{\pm 0.09}$ | $447.17_{\pm 4.19}$ | $472.08_{\pm 4.11}$ |
| | 1000 | TS2Vec | – | $128.12_{\pm 2.39}$ | $128.12_{\pm 2.39}$ |
| | | PLanTS | – | $387.98_{\pm 0.13}$ | $387.98_{\pm 0.13}$ |
| | | SoftCLT | $99.31_{\pm 0.43}$ | $899.07_{\pm 13.02}$ | $998.39_{\pm 12.94}$ |
| $C$ | 3 | TS2Vec | – | $57.55_{\pm 0.81}$ | $57.55_{\pm 0.81}$ |
| | | PLanTS | – | $166.26_{\pm 0.27}$ | $166.26_{\pm 0.27}$ |
| | | SoftCLT | $24.90_{\pm 0.09}$ | $447.17_{\pm 4.19}$ | $472.08_{\pm 4.11}$ |
| | 10 | TS2Vec | – | $57.44_{\pm 4.68}$ | $57.44_{\pm 4.68}$ |
| | | PLanTS | – | $170.59_{\pm 0.53}$ | $170.59_{\pm 0.53}$ |
| | | SoftCLT | $40.95_{\pm 0.39}$ | $448.41_{\pm 7.56}$ | $489.36_{\pm 7.93}$ |
| | 20 | TS2Vec | – | $58.24_{\pm 5.07}$ | $58.24_{\pm 5.07}$ |
| | | PLanTS | – | $178.64_{\pm 0.58}$ | $178.64_{\pm 0.58}$ |
| | | SoftCLT | $61.11_{\pm 0.55}$ | $456.37_{\pm 10.39}$ | $517.48_{\pm 10.95}$ |

phase. For example, at $L = 256$, SoftCLT requires over 715 seconds in total—driven largely by an expensive DTW-based precomputation step ($380.78 \pm 2.52$ seconds)—whereas PLanTS completes training in $110.93 \pm 0.61$ seconds. This advantage is maintained for longer sequences: at $L = 1024$, SoftCLT takes $650.23 \pm 2.57$ seconds, while PLanTS requires only $234.07 \pm 0.37$ seconds. Similar trends are observed when scaling the number of samples or channels, confirming the scalability and computational efficiency of PLanTS.

# F ANOMALY DETECTION TASK

We preform point-based anomaly detection experiment on Yahoo datasetRen et al. (2019). We follow the evaluation protocol of Yue et al. (2022). Given time series slice $x_1, x_2, ..., x_t$, the target is to determine whether the last time point $x_t$ is an anomaly. The anomaly score is computed as the $L_1$ distance between representations with masked and unmasked input. We evaluate PLanTS under two experiment setting: normal setting and cold-start setting, and compare results against 11 baseline methods. For normal setting, we consider SPOT, DSPOT ,DONUT and SR. For cold-start setting, we compare wtih FFT, Twitter-AD, Luminol and SR. We also use SSL methods:TS2Vec, T-Rep and SoftCLT as baseline methods for both settings. The results are reported in Table 8. From the results, PLanTS outperforms all the baseline methods in terms of F1 score. Remarkably, PLanTS improves F1 score approximately 2% with respect to SoftCLT and TS2Vec.

Table 8: Time series anomaly detection results.

| Method | Yahoo Normal | | | Method | Yahoo Cold Start | | |
|---|---|---|---|---|---|---|---|
| | F1 | Prec | Rec | | F1 | Prec | Rec |
| SPOT | 33.8 | 26.9 | 45.4 | FFT | 29.1 | 20.2 | 51.7 |
| DSPOT | 31.6 | 24.1 | 45.8 | Twitter-AD | 24.5 | 16.6 | 46.2 |
| DONUT | 2.6 | 1.3 | 82.5 | Luminol | 38.8 | 25.4 | 81.8 |
| SR | 5.63 | 45.1 | 74.7 | SR | 52.9 | 40.4 | 76.5 |
| TS2Vec | 74.5 | 72.9 | 76.2 | TS2Vec | 72.6 | 69.2 | 76.3 |
| T-Rep | 75.7 | 81.0 | 74.5 | T-Rep | 76.3 | 79.4 | 73.4 |
| SoftCLT | 74.2 | 72.2 | 76.5 | SoftCLT | 76.2 | 75.3 | 77.3 |
| PLanTS | **77.3** | 84.1 | 71.5 | PLanTS | **77.4** | 83.7 | 72.0 |

Table 9: Full classification results on 30 UEA datasets.

| Dataset | PLanTS | CLS | softclt | T-Rep | TS2Vec | T-Loss | TNC | TS-TCC | TST | DTW |
|---|---|---|---|---|---|---|---|---|---|---|
| ArticularyWordRecognition | 0.973 | 0.943 | 0.990 | 0.957 | 0.943 | 0.943 | 0.973 | 0.953 | 0.977 | 0.987 |
| AtrialFibrillation | 0.267 | 0.467 | 0.200 | 0.267 | 0.133 | 0.133 | 0.133 | 0.267 | 0.067 | 0.200 |
| BasicMotions | 1.000 | 0.975 | 0.975 | 1.000 | 0.975 | 1.000 | 0.975 | 1.000 | 0.975 | 0.975 |
| CharacterTrajectories | 0.983 | 0.985 | 0.992 | 0.983 | 0.987 | 0.993 | 0.967 | 0.985 | 0.975 | 0.989 |
| Cricket | 1.000 | 0.944 | 0.972 | 0.972 | 0.972 | 0.972 | 0.958 | 0.917 | 1.000 | 1.000 |
| DuckDuckGeese | 0.560 | 0.440 | 0.360 | 0.457 | 0.680 | 0.650 | 0.460 | 0.380 | 0.620 | 0.600 |
| EigenWorms | 0.809 | 0.884 | – | 0.884 | 0.847 | 0.840 | 0.840 | 0.779 | 0.748 | 0.618 |
| Epilepsy | 0.971 | 0.970 | 0.942 | 0.970 | 0.964 | 0.971 | 0.957 | 0.957 | 0.949 | 0.964 |
| ERing | 0.852 | 0.943 | 0.941 | 0.943 | 0.874 | 0.133 | 0.852 | 0.904 | 0.874 | 0.133 |
| EthanolConcentration | 0.274 | 0.264 | 0.278 | 0.333 | 0.308 | 0.205 | 0.297 | 0.285 | 0.262 | 0.323 |
| FaceDetection | 0.550 | 0.548 | 0.493 | 0.581 | 0.501 | 0.513 | 0.536 | 0.544 | 0.534 | 0.529 |
| FingerMovements | 0.580 | 0.540 | 0.580 | 0.495 | 0.480 | 0.580 | 0.470 | 0.460 | 0.560 | 0.530 |
| HandMovementDirection | 0.446 | 0.473 | 0.392 | 0.536 | 0.338 | 0.351 | 0.324 | 0.243 | 0.243 | 0.231 |
| Handwriting | 0.439 | 0.343 | 0.467 | 0.414 | 0.515 | 0.451 | 0.249 | 0.498 | 0.225 | 0.286 |
| Heartbeat | 0.746 | 0.742 | 0.722 | 0.725 | 0.683 | 0.741 | 0.746 | 0.751 | 0.746 | 0.717 |
| JapaneseVowels | 0.976 | 0.942 | 0.978 | 0.962 | 0.984 | 0.989 | 0.978 | 0.930 | 0.978 | 0.949 |
| Libras | 0.861 | 0.844 | 0.889 | 0.829 | 0.867 | 0.883 | 0.817 | 0.822 | 0.656 | 0.870 |
| LSST | 0.598 | 0.526 | 0.534 | 0.526 | 0.537 | 0.509 | 0.595 | 0.474 | 0.408 | 0.551 |
| MotorImagery | 0.570 | 0.525 | – | 0.495 | 0.510 | 0.580 | 0.500 | 0.610 | 0.500 | 0.500 |
| NATOPS | 0.917 | 0.854 | 0.944 | 0.804 | 0.928 | 0.917 | 0.911 | 0.822 | 0.850 | 0.883 |
| PEMS-SF | 0.803 | 0.813 | 0.723 | 0.800 | 0.682 | 0.675 | 0.699 | 0.734 | 0.740 | 0.711 |
| PenDigits | 0.986 | 0.983 | 0.987 | 0.971 | 0.989 | 0.981 | 0.979 | 0.974 | 0.560 | 0.977 |
| PhonemeSpectra | 0.247 | 0.257 | 0.223 | 0.232 | 0.233 | 0.222 | 0.207 | 0.252 | 0.085 | 0.151 |
| RacketSports | 0.842 | 0.866 | 0.855 | 0.883 | 0.855 | 0.855 | 0.776 | 0.816 | 0.809 | 0.803 |
| SelfRegulationSCP1 | 0.901 | 0.847 | 0.799 | 0.819 | 0.812 | 0.843 | 0.799 | 0.823 | 0.754 | 0.775 |
| SelfRegulationSCP2 | 0.544 | 0.572 | 0.500 | 0.591 | 0.578 | 0.539 | 0.550 | 0.533 | 0.550 | 0.539 |
| SpokenArabicDigits | 0.951 | 0.961 | 0.949 | 0.994 | 0.988 | 0.905 | 0.934 | 0.970 | 0.923 | 0.963 |
| StandWalkJump | 0.667 | 0.567 | 0.533 | 0.441 | 0.467 | 0.332 | 0.400 | 0.333 | 0.267 | 0.200 |
| UWaveGestureLibrary | 0.850 | 0.828 | 0.925 | 0.885 | 0.906 | 0.875 | 0.759 | 0.753 | 0.575 | 0.903 |
| InsectWingbeat | 0.423 | 0.363 | – | 0.328 | 0.466 | 0.156 | 0.469 | 0.264 | 0.105 | – |
| Avg. Acc. | **0.720** | 0.698 | 0.709 | 0.693 | 0.690 | 0.670 | 0.658 | 0.668 | 0.617 | 0.650 |
| Avg. Rank | **3.333** | 4.867 | 4.481 | 4.667 | 5.100 | 4.833 | 5.767 | 5.633 | 6.900 | 5.862 |
| Ranks 1st | 8 | 7 | 5 | 4 | 3 | 1 | 2 | 4 | 2 | 1 |
| W/T/L | – | 18/0/12 | 16/1/10 | 18/5/7 | 21/0/9 | 17/4/9 | 23/3/4 | 20/2/8 | 24/2/4 | 21/1/7 |
| p-value | – | 0.089 | 0.033 | 0.036 | 0.003 | 0.009 | 0.000 | 0.001 | 0.000 | 0.001 |

## G  FULL RESULTS

The full results of MTS classification task on 30 UEA datasets are shown in Table 9. The full results for the forecasting task on the 4 ETT datasets are presented in Table 10. Figure 7 and Figure 8 shows the per-class AUROC for 10 selected form categories and 10 selected rhythm categories, respectively. For trajectory tracking task, Figure 9 shows another example of comparison among top 3 principal components (PCA) of the learned embeddings of PLanTS, TS2Vec and SoftCLT.

## H  LLM USAGE

This paper used a Large Language Model (OpenAI ChatGPT) as a general-purpose writing assistant. The LLM was employed for: (i) polishing grammar, and (ii) suggesting LaTeX formatting for equations and tables.

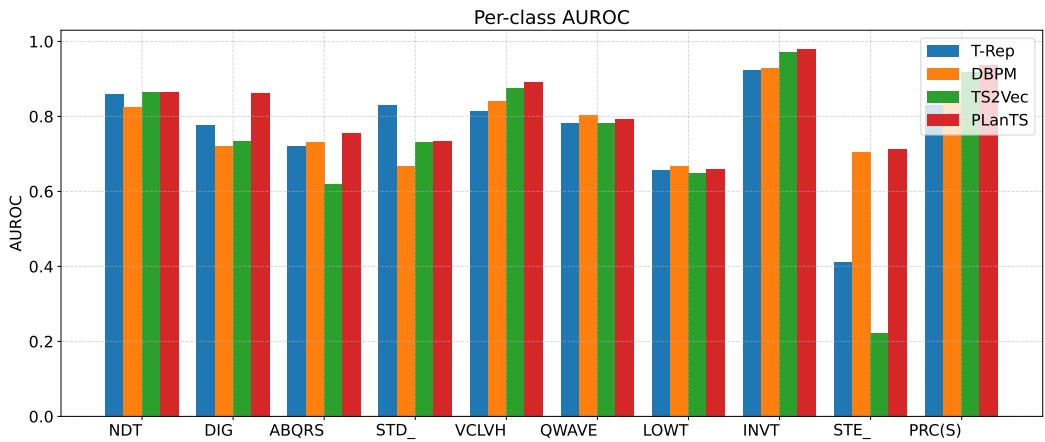

Figure 7: Per-class AUROC comparison on 10 selected form classes from PTB-XL.

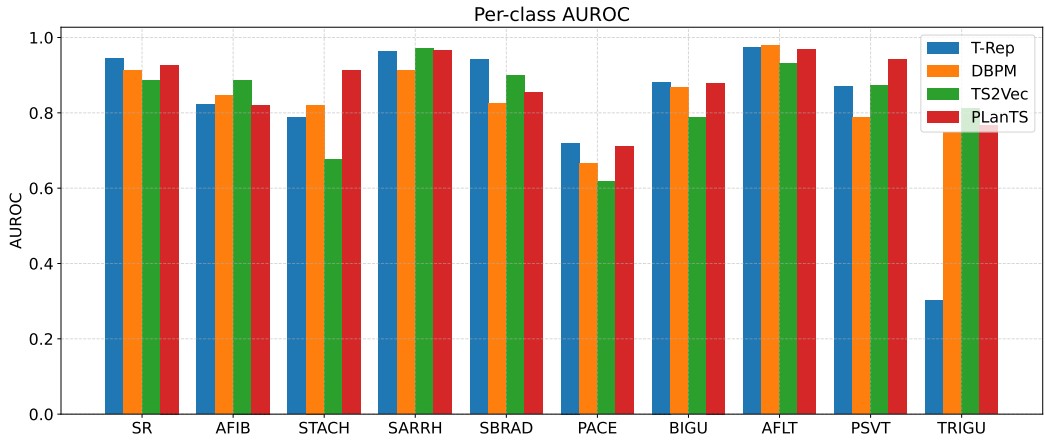

Figure 8: Per-class AUROC comparison on 10 selected rhythm classes from PTB-XL.

The LLM was not involved in research ideation, algorithm design, experiment implementation, or result analysis. All technical contributions, models, experiments, and conclusions were conceived, implemented, and validated solely by the authors.

Table 10: Forecasting results on ETT datasets across multiple horizons.

| Dataset | H | PLanTS MSE | PLanTS MAE | SoftClt MSE | SoftClt MAE | T-rep MSE | T-rep MAE | TS2Vec MSE | TS2Vec MAE | Informer MSE | Informer MAE | TCN MSE | TCN MAE |
|---|---|---|---|---|---|---|---|---|---|---|---|---|---|
| ETTh1 | 24 | 0.518 | 0.508 | 0.630 | 0.550 | **0.511** | **0.496** | 0.575 | 0.529 | 0.577 | 0.549 | 0.767 | 0.612 |
|  | 48 | 0.547 | 0.529 | 0.670 | 0.579 | **0.546** | **0.524** | 0.608 | 0.553 | 0.685 | 0.625 | 0.713 | 0.617 |
|  | 168 | **0.676** | **0.607** | 0.814 | 0.664 | 0.759 | 0.649 | 0.782 | 0.659 | 0.931 | 0.752 | 0.995 | 0.738 |
|  | 336 | **0.827** | **0.687** | 0.976 | 0.749 | 0.936 | 0.742 | 0.956 | 0.753 | 1.128 | 0.873 | 1.175 | 0.800 |
|  | 720 | **0.971** | **0.773** | 1.088 | 0.807 | 1.061 | 0.813 | 1.092 | 0.831 | 1.215 | 0.896 | 1.453 | 1.311 |
| ETTh2 | 24 | **0.364** | **0.443** | 0.384 | 0.458 | 0.560 | 0.565 | 0.448 | 0.506 | 0.720 | 0.665 | 1.365 | 0.888 |
|  | 48 | 0.630 | 0.603 | **0.55** | **0.564** | 0.847 | 0.711 | 0.685 | 0.642 | 1.457 | 1.001 | 1.395 | 0.960 |
|  | 168 | 2.167 | 1.137 | **1.722** | **1.026** | 2.327 | 1.206 | 2.227 | 1.164 | 3.489 | 1.515 | 3.166 | 1.407 |
|  | 336 | 2.641 | 1.303 | **2.174** | **1.193** | 2.665 | 1.324 | 2.803 | 1.360 | 2.723 | 1.340 | 3.256 | 1.481 |
|  | 720 | **2.623** | **1.349** | 2.642 | 1.383 | 2.690 | 1.365 | 2.849 | 1.436 | 3.467 | 1.473 | 3.690 | 1.588 |
| ETTm1 | 24 | 0.370 | 0.398 | 0.453 | 0.445 | 0.417 | 0.420 | 0.438 | 0.435 | **0.323** | **0.369** | 0.324 | 0.374 |
|  | 48 | 0.485 | 0.472 | 0.604 | 0.523 | 0.526 | 0.484 | 0.582 | 0.555 | 0.494 | 0.505 | **0.477** | **0.450** |
|  | 96 | **0.526** | **0.501** | 0.622 | 0.537 | 0.573 | 0.516 | 0.602 | 0.537 | 0.678 | 0.614 | 0.636 | 0.602 |
|  | 288 | **0.590** | **0.551** | 0.686 | 0.586 | 0.648 | 0.577 | 0.709 | 0.610 | 1.056 | 0.786 | 1.270 | 1.351 |
|  | 672 | **0.684** | **0.612** | 0.774 | 0.644 | 0.758 | 0.649 | 0.826 | 0.687 | 1.192 | 0.926 | 1.381 | 1.467 |
| ETTm2 | 24 | **0.129** | **0.244** | 0.173 | 0.293 | 0.172 | 0.293 | 0.189 | 0.310 | 0.147 | 0.277 | 1.452 | 1.938 |
|  | 48 | **0.189** | **0.304** | 0.253 | 0.362 | 0.263 | 0.377 | 0.256 | 0.369 | 0.267 | 0.389 | 2.181 | 0.839 |
|  | 96 | **0.270** | **0.375** | 0.371 | 0.446 | 0.397 | 0.470 | 0.402 | 0.471 | 0.317 | 0.411 | 3.921 | 1.714 |
|  | 288 | 0.783 | **0.656** | **0.728** | 0.662 | 0.897 | 0.733 | 0.879 | 0.724 | 1.147 | 0.834 | 3.649 | 3.245 |
|  | 672 | 3.053 | 1.328 | **1.702** | **1.144** | 2.185 | 1.144 | 2.193 | 1.159 | 3.989 | 1.598 | 6.973 | 1.719 |
| Weather | 96 | 0.196 | 0.279 | 0.206 | 0.287 | 0.203 | 0.289 | **0.138** | **0.213** | 0.300 | 0.384 | 0.615 | 0.589 |
|  | 192 | **0.237** | **0.316** | 0.250 | 0.326 | 0.252 | 0.330 | 0.362 | 0.378 | 0.598 | 0.544 | 0.629 | 0.600 |
|  | 336 | **0.298** | **0.364** | 0.305 | 0.368 | 0.310 | 0.373 | 0.653 | 0.528 | 0.578 | 0.523 | 0.639 | 0.608 |
|  | 720 | **0.317** | **0.411** | 0.378 | 0.416 | 0.383 | 0.419 | 0.935 | 0.674 | 1.059 | 0.741 | 0.639 | 0.610 |
| Avg. Rank |  | **1.75** | **1.75** | 2.96 | 2.96 | 2.88 | 2.88 | 3.92 | 3.92 | 4.54 | 4.54 | 5.96 | 5.96 |
| Ranks 1st |  | **13** | **13** | 5 | 5 | 2 | 2 | 1 | 1 | 1 | 1 | 1 | 1 |

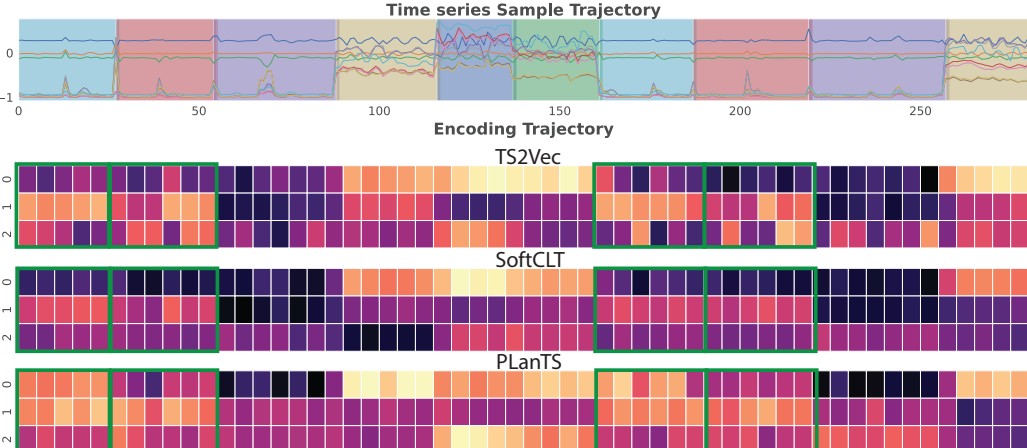

Figure 9: Trajectory of another HAR signal encoding.

