# OpenReview forum: "PLanTS: Periodicity-aware Latent-state Representation Learning for Multivariate Time Series"
_ICLR.cc/2026/Conference — Submitted to ICLR 2026_

### Official Review · Reviewer_us21 · 2025-10-23

**Soundness:** 2
**Presentation:** 2
**Contribution:** 1
**Rating:** 4
**Confidence:** 4

**Summary:**

This paper introduces PLanTS, a periodicity-aware self-supervised learning framework for multivariate time series. It models latent states and their transitions using a periodicity-aware multi-granularity patching mechanism and a contrastive loss combining instance and state-level samples. Additionally, it includes a next-transition prediction task to capture temporal dynamics.

**Strengths:**

The next transition task is novel in time series. The paper is also easy to read and follow. Authors also performed several experiments including several tasks, such as classification and forecasting.

**Weaknesses:**

The idea of multi-granularity patching is not new, and using FFT to extract dominant periodicities for patching has already been explored, for example in TimesNet. My main concern is the next-transition prediction task.
Since the PTB-XL dataset consists of 10-second, 12-lead ECG samples, it is unclear how the model can meaningfully learn state transitions within such short segments. This approach seems more suitable for continuous recordings with well-defined transitions. Also, most of the datasets are collected in a controlled environments where the next state transition is already set. How does the model can deal with that situation?

**Questions:**

1) How does the proposed multi-granularity patching differ from prior FFT-based methods like TimesNet?

2) With only 10-second PTB-XL samples, what does “next-state transition” mean, and how can the model learn it without continuous data?

3) Has the next-transition task been tested on datasets with longer recordings to verify its validity?

---

> ### Author Response · Authors · 2025-11-20
>
> Dear reviewer us21,
>
> We appreciate your time and effort in reviewing our manuscript. Here are our responses and adjustments according to the reviewer's comments.
>
> Response to: The idea of multi-granularity patching is not new, and using FFT to extract dominant periodicities for patching has already been explored, for example in TimesNet.
>
> We acknowledge that FFT-based periodicity detection has been explored previously (e.g., TimesNet). However, our contribution differs in how the periodic information is used. Instead of directly learning 2D variations using 2D convolutional kernels as in TimesNet, PLanTS leverages FFT to determine adaptive window sizes for contrastive segmentation, enabling hierarchical contrastive learning across multiple latent temporal scales. This design bridges periodicity detection and latent state modeling in a unified self-supervised framework, which to our knowledge has not been explored before.
>
> Response to: My main concern is the next-transition prediction task. Since the PTB-XL dataset consists of 10-second, 12-lead ECG samples, it is unclear how the model can meaningfully learn state transitions within such short segments. This approach seems more suitable for continuous recordings with well-defined transitions. Also, most of the datasets are collected in a controlled environments where the next state transition is already set. How does the model can deal with that situation?
>
> We thank the reviewer for the insightful comments regarding the applicability of the next-transition prediction task. We would like to clarify that in the PTB-XL experiments, we set the hyperparameter $\lambda = 1$ (see Appendix B implementation details and Appendix Table 6: Hyperparameter settings for various tasks), disabling the next-transition prediction loss in the total objective. We agree that: PTB-XL samples are short (10 s) and do not exhibit meaningful latent transitions within each recording, making the dynamic transition pretext task less informative.
>
> The Dynamic Transition Module in PLanTS is designed primarily for continuous or long-horizon datasets (e.g., Human Activity Recognition, ETT forecasting, industrial sensor monitoring), where the underlying latent states evolve gradually over time. In contrast, for short or isolated recordings where temporal continuity is limited, the transition modeling component can be safely omitted without affecting performance.
>
> We will add discussion in section 4.2 to clarify this design choice and explicitly state the conditions under which the dynamic transition loss is active or omitted:
>
> When does dynamic transition modeling apply?
>
> The next-transition prediction pretext task is designed for datasets that exhibit continuous temporal dynamics, where latent states evolve gradually over time—such as Human Activity Recognition and long-horizon forecasting (ETT). In these settings, modeling transitions between adjacent latent states helps capture smooth temporal evolution and predictive structure.
>
> However, for datasets consisting of short, isolated sequences without meaningful intra-record transitions (e.g., PTB-XL, where each 10 s ECG record is largely stationary), the transition objective contributes little additional information. For such datasets, we set $\lambda = 1$ to disable the dynamic transition loss and focus on the latent-state representation objective. This adaptive usage ensures PLanTS remains effective across both dynamic and static time-series domains.
>
> Responses to question 1 and 2: Please prefer to our responses above.
>
> Response to question 3: Has the next-transition task been tested on datasets with longer recordings to verify its validity?
>
> Thanks to the reviewer for pointing out this question. We did ablation and tested how next-transition task influence the forecasting results of longer recordings datasets, like 4 ETT datasets. Please see details in Appendix C Hyper-parameter sensitivity. Hyper-parameter $\lambda$ control the relative weights of  latent state contrastive loss and next-transition task loss(larger $\lambda$ indicate smaller weight of next-transition task loss). Appendix Figure 6 reports the percentage change of MSE and MAE with respect to the best results across four datasets.  We observed that: with the increase of $\lambda$ (decrease weight of next-transition task loss), the MSE and MAE increase particularly on ETTh2. Suggesting that next-transition task is critical for datasets with more complex temporal dependencies.

---

### Official Review · Reviewer_rZwx · 2025-10-26

**Soundness:** 2
**Presentation:** 2
**Contribution:** 2
**Rating:** 4
**Confidence:** 5

**Summary:**

This paper focus on the self-supervised learning for multivariate time series modeling. A model named PLanTS is proposed, which is a periodicity-aware self-supervised learning framework that explicitly models irregular latent states and their transitions. Key techniques include multi-granularity patching and contrastive. Experiments are carried out based on classification, forecasting, trajectory tracking, and anomaly detection tasks.

**Strengths:**

1. SSL for MTS modeling is a critical problem

2. Source codes are provided to ensure reproducibility.

**Weaknesses:**

1. More discussions on the motivation are needed. Why do we need periodicity-aware and multi-granularity self-supervised learning for MTS modeling? Under what challenges and problems?

2. Does multi-granularity patching relate to multi-scale modeling? There are many related works that need discussion, e.g., Timemixer.

3. Experiments need improvement; more baseline methods could be included, e.g., methods that are based on masked-reconstruction (SimMTM), and TimeSiam.

4. To evaluate the time series forecasting performance. More datasets except for ETTs are needed.

**Questions:**

Please refer to the weakness

---

> ### Author Response · Authors · 2025-12-02
>
> Dear Reviewer  rZwx,
>
> We appreciate your time and effort in reviewing our manuscript. Here are our responses and adjustments according to the reviewer's comments.
>
> Response to weakness 1:
>
> We thank the reviewer for pointing out the need for more explicit motivation.
>
> MTS data frequently exhibit quasi-periodic structure, nonstationarity, and multi-scale temporal dependencies, which existing SSL methods do not address. Point-based or fixed-window contrastive learning assumes consistent temporal dynamics and overlooks periodic patterns, leading to incorrect similarity assignments. Moreover, latent states evolve across different temporal horizons (e.g., short-term fluctuations vs. long-term transitions), making single-scale modeling insufficient.
>
> Our periodicity-aware, multi-granularity design directly targets these challenges: FFT-based periodicity extraction aligns patching with the intrinsic rhythms of the signal, and multi-scale patches capture latent states and transitions at heterogeneous temporal resolutions. In detail,  each scale captured by FFT corresponds to one granularity, which is then processed independently by two encoders, capturing latent state structure and predictive information of future state transitions respectively. A novel generalized contrastive loss will be computed for the representation at each scale, and the final loss will be the average of all the scales.
> We will revise the introduction to explicitly highlight these motivations. See revised PDF line 61-68.
>
>
> Response to weakness 2:
>
> We thank the reviewer for raising this important point.
>
> Yes—our multi-granularity patching mechanism is conceptually related to multi-scale modeling, and we will clarify this in the revised paper.
> However, existing multi-scale approaches such as TimeMixer, TimesNet, and other hierarchical architectures differ from PLanTS in key aspects:
>
> 1. Purpose:
> Prior works use multi-scale architectures primarily for supervised forecasting, while PLanTS uses multi-granularity only during self-supervised representation learning. At test time, no patching is applied.
>
> 2. Model construction: Instead of directly learning 2D variations using 2D convolutional kernels as in TimesNet, PLanTS leverages FFT to determine adaptive window sizes for contrastive segmentation, enabling hierarchical contrastive learning across multiple latent temporal scales. This design bridges periodicity detection and latent state modeling in a unified self-supervised framework, which to our knowledge has not been explored before.
>
> In our method, multi-granularity patches correspond to different temporal receptive fields. Each granularity is processed independently by the latent-state and dynamic transition encoders, capturing latent state structure and predictive information of future state transitions respectively. A contrastive loss will be computed for the representation at each scale, and the final loss will be the average of all the scales.
>
> Response to weakness 3:
>
> This issue overlaps substantially with Reviewer tVjj’s Comment 1. To avoid redundancy, we kindly refer the reviewer to our detailed response to that comment, where we provide an in-depth explanation and supporting analysis.
>
> Response to weakness 4:
>
> Thanks to the reviewer for pointing out that more large-scale forecasting datasets could be included.  We additionally incorporated the Weather dataset [5](a widely used large-scale long-horizon forecasting benchmark). PLanTS was evaluated following the same experimental protocol described in Section 4.3.
>
> Please see the result table as we reply to  Reviewer tVjj’s Comment 3.

---

### Official Review · Reviewer_tVjj · 2025-10-29

**Soundness:** 2
**Presentation:** 2
**Contribution:** 2
**Rating:** 4
**Confidence:** 4

**Summary:**

In this paper, the authors propose a periodicity-aware, multi-granularity self-supervised learning framework for non-stationary multivariate time series representation learning based on contrastive learning. This framework can be applied to downstream classification, forecasting, trajectory tracking, and anomaly detection tasks.

**Strengths:**

1. SSL for MTS modeling is a critical task.

2. Source codes are provided.

3. The paper is well-structured and easy-to-follow

**Weaknesses:**

1. Some critical related works are missing. For example, multi-scale modeling methods for time-series like Timemixer++. Also, there are numerous time series foundation models that require pre-training based on large-scale datasets, which can also be regarded as representation learning for MTS.

2. It's unclear why multi-scale modeling can address nonstationary challenges in MTS.


3. More large-scale forecasting benchmarks could be included, e.g., weather and traffic datasets like PEMS.

**Questions:**

1. Since the proposed method adopts multi-scale modeling techniques. Can itbe  applied to multi-rate time series modeling? More discussions are needed.

---

> ### Author Response · Authors · 2025-12-02
>
> Dear ReviewertVjj,
>
> We appreciate your time and effort in reviewing our manuscript. Here are our responses and adjustments according to the reviewer's comments.
>
>
> Response to weakness 1:
>
> We appreciate the reviewer’s suggestion to include additional related works. We will incorporate discussion of methods such as TimeMixer++, which offer useful perspectives on multi-scale modeling. However, we would also like to clarify that a direct comparison with such models would be methodologically inappropriate due to fundamental differences in training paradigms.
> As highlighted in several published studies[1], existing time-series methods generally fall into two families:
>
> 1. Pretrain–Finetune Paradigm.
>
> Representative works include TimeMixer++, SimMTM, and others. For each dataset and downstream task, these approaches first pretrain a feature extractor, then design a task-specific head and finetune the entire network end-to-end. While this paradigm can achieve strong performance because of full end-to-end optimization, it is computationally expensive, requires task-specific finetuning of the backbone for every use case, and resembles training a separate model for each task rather than learning a
> general-purpose representation.
>
> 2. General Representation Learning Paradigm.
>
> In this setting, include works like TS2Vec(2023)[2], T-rep(2024)[3], softCLT(2024)[4] and PLanTS(our method),  a representation encoder is trained once for a dataset, and its representations are reused for downstream tasks without finetuning the encoder. Only lightweight downstream models are trained (SVM for classification and ridge regression for forecasting). Our method and all baselines evaluated in this manuscript belong to this family. Because of this difference in paradigm, direct comparison with pretrain–finetune methods is not meaningful or fair.
>
> We also want to discuss the impact of backbone architecture on performance. Our contribution focuses on the representation learning strategy rather than proposing a new backbone. When compared with representation-learning methods using the same convolutional backbone (TS2Vec, T-rep and softCLT), our method consistently performs better. Recent approaches such as TimeMixer++ employ Transformer-based backbones, which may yield stronger performance independently of training strategy. We believe our proposed strategy could similarly benefit Transformer-based encoders, but conducting extensive analyses with alternative backbones is beyond the scope of the present work.
>
> [1]Trirat, Patara, et al. "Universal time-series representation learning: A survey." arXiv preprint arXiv:2401.03717 (2024).
>
> [2]Yue, Zhihan, et al. "Ts2vec: Towards universal representation of time series." Proceedings of the AAAI conference on artificial intelligence. Vol. 36. No. 8. 2022.
>
> [3]Fraikin, Archibald, Adrien Bennetot, and Stéphanie Allassonnière. "T-Rep: Representation Learning for Time Series using Time-Embeddings." The Twelfth International Conference on Learning Representations. 2024.
>
> [4]Lee, Seunghan, Taeyoung Park, and Kibok Lee. "SOFT CONTRASTIVE LEARNING FOR TIME SERIES." 12th International Conference on Learning Representations, ICLR 2024. 2024.
>
> Response to weakness 2:
>
> We thank the reviewer for raising this important point. We agree that the connection between multi-scale modeling and handling nonstationarity should be stated more clearly.
>
> In multivariate time series (MTS), nonstationarity arises because temporal dependencies and feature correlations vary across different time horizons. Multi-scale modeling provides a way to mitigate this by capturing both short-term and long-term temporal patterns. In PLanTS, the periodicity-aware multi-granularity patching mechanism adaptively decomposes the input sequence into multiple temporal resolutions guided by dominant periodicities. Each granularity is processed independently by the latent-state and dynamic transition encoders, capturing latent state structure and predictive information of future state transitions respectively.
>  This enables the model to: (1) learn local, fine-grained representations that reflect transient behaviors, and (2) learn coarse-grained patterns representing slow, structural trends.
>
>  By minimizing the contrastive loss computed for the representation at each scale, PLanTS becomes more robust to shifts in statistical properties over time, which effectively addresses nonstationarity.
>
> We will clarify this connection in the revised manuscript by adding explanation in Section 3.2.

---

> ### Author Response · Authors · 2025-12-02
>
> Response to weakness 3:
>
> Thanks to the reviewer for pointing out that more large-scale forecasting datasets could be included. To further strengthen the evaluation of PLanTS on real-world forecasting tasks, we additionally incorporated the Weather dataset [5](a widely used large-scale long-horizon forecasting benchmark). PLanTS was evaluated following the same experimental protocol described in Section 4.3.
>
> The results are presented below, and they consistently show that PLanTS maintains strong forecasting performance compared with existing self-supervised baselines. This additional experiment further demonstrates the robustness and generalization ability of PLanTS on forecasting tasks.
>
> Table: Forecasting results on ETT and Weather datasets across multiple horizons.
>
> | Dataset | H | PLanTS (MSE) | PLanTS (MAE) | SoftClt (MSE) | SoftClt (MAE) | T-rep (MSE) | T-rep (MAE) | TS2Vec (MSE) | TS2Vec (MAE) | Informer (MSE) | Informer (MAE) | TCN (MSE) | TCN (MAE) |
> |--------|----|---------------|---------------|----------------|----------------|-------------|-------------|---------------|---------------|------------------|------------------|-------------|-------------|
> | **ETTh1** | 24  | 0.518 | 0.508 | 0.630 | 0.550 | **0.511** | **0.496** | 0.575 | 0.529 | 0.577 | 0.549 | 0.767 | 0.612 |
> |        | 48  | 0.547 | 0.529 | 0.670 | 0.579 | **0.546** | **0.524** | 0.608 | 0.553 | 0.685 | 0.625 | 0.713 | 0.617 |
> |        | 168 | **0.676** | **0.607** | 0.814 | 0.664 | 0.759 | 0.649 | 0.782 | 0.659 | 0.931 | 0.752 | 0.995 | 0.738 |
> |        | 336 | **0.827** | **0.687** | 0.976 | 0.749 | 0.936 | 0.742 | 0.956 | 0.753 | 1.128 | 0.873 | 1.175 | 0.800 |
> |        | 720 | **0.971** | **0.773** | 1.088 | 0.807 | 1.061 | 0.813 | 1.092 | 0.831 | 1.215 | 0.896 | 1.453 | 1.311 |
> | **ETTh2** | 24  | **0.364** | **0.443** | 0.384 | 0.458 | 0.560 | 0.565 | 0.448 | 0.506 | 0.720 | 0.665 | 1.365 | 0.888 |
> |        | 48  | 0.630 | 0.603 | **0.550** | **0.564** | 0.847 | 0.711 | 0.685 | 0.642 | 1.457 | 1.001 | 1.395 | 0.960 |
> |        | 168 | 2.167 | 1.137 | **1.722** | **1.026** | 2.327 | 1.206 | 2.227 | 1.164 | 3.489 | 1.515 | 3.166 | 1.407 |
> |        | 336 | 2.641 | 1.303 | **2.174** | **1.193** | 2.665 | 1.324 | 2.803 | 1.360 | 2.723 | 1.340 | 3.256 | 1.481 |
> |        | 720 | **2.623** | **1.349** | 2.642 | 1.383 | 2.690 | 1.365 | 2.849 | 1.436 | 3.467 | 1.473 | 3.690 | 1.588 |
> | **ETTm1** | 24  | 0.370 | 0.398 | 0.453 | 0.445 | 0.417 | 0.420 | 0.438 | 0.435 | **0.323** | **0.369** | 0.324 | 0.374 |
> |        | 48  | 0.485 | 0.472 | 0.604 | 0.523 | 0.526 | 0.484 | 0.582 | 0.555 | 0.494 | 0.505 | **0.477** | **0.450** |
> |        | 96  | **0.526** | **0.501** | 0.622 | 0.537 | 0.573 | 0.516 | 0.602 | 0.537 | 0.678 | 0.614 | 0.636 | 0.602 |
> |        | 288 | **0.590** | **0.551** | 0.686 | 0.586 | 0.648 | 0.577 | 0.709 | 0.610 | 1.056 | 0.786 | 1.270 | 1.351 |
> |        | 672 | **0.684** | **0.612** | 0.774 | 0.644 | 0.758 | 0.649 | 0.826 | 0.687 | 1.192 | 0.926 | 1.381 | 1.467 |
> | **ETTm2** | 24  | **0.129** | **0.244** | 0.173 | 0.293 | 0.172 | 0.293 | 0.189 | 0.310 | 0.147 | 0.277 | 1.452 | 1.938 |
> |        | 48  | **0.189** | **0.304** | 0.253 | 0.362 | 0.263 | 0.377 | 0.256 | 0.369 | 0.267 | 0.389 | 2.181 | 0.839 |
> |        | 96  | **0.270** | **0.375** | 0.371 | 0.446 | 0.397 | 0.470 | 0.402 | 0.471 | 0.317 | 0.411 | 3.921 | 1.714 |
> |        | 288 | 0.783 | **0.656** | **0.728** | 0.662 | 0.897 | 0.733 | 0.879 | 0.724 | 1.147 | 0.834 | 3.649 | 3.245 |
> |        | 672 | 3.053 | 1.328 | **1.702** | **1.144** | 2.185 | 1.144 | 2.193 | 1.159 | 3.989 | 1.598 | 6.973 | 1.719 |
> | **Weather** | 96  | 0.196 | 0.279 | 0.206 | 0.287 | 0.203 | 0.289 | **0.138** | **0.213** | 0.300 | 0.384 | 0.615 | 0.589 |
> |        | 192 | **0.237** | **0.316** | 0.250 | 0.326 | 0.252 | 0.330 | 0.362 | 0.378 | 0.598 | 0.544 | 0.629 | 0.600 |
> |        | 336 | **0.298** | **0.364** | 0.305 | 0.368 | 0.310 | 0.373 | 0.653 | 0.528 | 0.578 | 0.523 | 0.639 | 0.608 |
> |        | 720 | **0.317** | **0.411** | 0.378 | 0.416 | 0.383 | 0.419 | 0.935 | 0.674 | 1.059 | 0.741 | 0.639 | 0.610 |
> | **Avg. Rank** | — | **1.75** | **1.75** | 2.96 | 2.96 | 2.88 | 2.88 | 3.92 | 3.92 | 4.54 | 4.54 | 5.96 | 5.96 |
> | **Ranks 1st** | — | **13** | **13** | 5 | 5 | 2 | 2 | 1 | 1 | 1 | 1 | 1 | 1 |
>
>  [5]Wetterstation. Weather. https://www.bgc-jena.mpg.de/wetter/.

---

### Official Review · Reviewer_oiHR · 2025-11-01

**Soundness:** 2
**Presentation:** 2
**Contribution:** 2
**Rating:** 4
**Confidence:** 5

**Summary:**

This paper proposed a representation learning methods for multi-variate time series data. The method first uses Fourier transformation to find frequency components with top-k amplitudes to select window sizes to perform multi-scale time series patching. Then the representations for the similar subsequence patched among patches in one times series of different start time stamp or across different time series with the same time stamp are aligned according to “Maximum Cross Correlation” scores using InfoNCE loss. The two parts of loss are used to optimize two different encoders and the representations learned is concatenated into a whole.

**Strengths:**

1. This paper proposes simple methods that is easy to follow.
2. The multiple types of downstream tasks enhance validity of proposed method.
3. The proposed method shows superiority over the selected baselines.

**Weaknesses:**

1. The methods are introduced in a straight way, which lack of salient analysis, insights ,and takeaways. Such as insights about the use of “Maximum Cross Correlation”, or how to feed the multi-scale patches into the encoders.
2. The baselines are pretty old, new methods should be presented, such as CSL[1].
3. Figure 2 introduces pretext tasks for TNC and TS2Vec. However the purpose for demonstrating them, e.g., for contrastive usage, is not introduced explicitly enough.
4. Taking an average over performances on 30 UEA dataset is not a reasonable operation, because there is no statistical meaning.
[1] Liang, Z., Zhang, J., Liang, C., Wang, H., Liang, Z., & Pan, L. (2023). A Shapelet-based Framework for Unsupervised Multivariate Time Series Representation Learning. Proc. VLDB Endow., 17, 386-399.

**Questions:**

As the weaknesses shows.

---

> ### Author Response · Authors · 2025-11-20
>
> Dear Reviewer oiHR,
> We appreciate your time and effort in reviewing our manuscript. Here are our responses and adjustments according to the reviewer's comments.
>
> Response to weakness 1: The methods are introduced in a straight way, which lack of salient analysis, insights ,and takeaways. Such as insights about the use of “Maximum Cross Correlation”, or how to feed the multi-scale patches into the encoders.
>
> We appreciate the reviewer for the valuable suggestion. We will include deeper insight and expand methodological discussion with respect to the two points that review mentioned:
>
>
> a. On Maximum Cross-Correlation (MXCorr):
>
>
> We will clarify that MXCorr is selected to efficiently measure temporal similarity between two windows without explicit alignment, unlike DTW. The key insight is that many multivariate time series exhibit phase-shifted periodic signals—MXCorr naturally aligns such signals in the frequency domain, providing a fast similarity measure suitable for contrastive learning. This allows PLanTS to model latent state structures in raw data space while avoiding expensive pairwise alignment.
>
>
> To clarify this, we will add the following paragraph in section 3.3 Latent State Representation (we will share more details on the position of this updated paragraph after we submit the new pdf):
>
>
> “Unlike DTW, which explicitly aligns sequences through dynamic programming, MXCorr measures the phase-invariant similarity between windows by finding the maximal normalized cross-correlation across possible temporal shifts. This property is particularly beneficial for quasi-periodic signals (e.g., ECG, sensor motion), where latent states may exhibit small phase shifts. By leveraging feature similarities captured through MXCorr, PLanTS effectively preserves latent state structures directly in the raw data space, providing informative self-supervision for robust latent-state representation learning.”
>
>
> b. On multi-scale patch encoding:
>
> We will add an explanation that multi-granularity patches correspond to different temporal receptive fields. Each granularity is processed independently by the latent-state and dynamic transition encoders, capturing latent state structure and predictive information of future state transitions respectively. A contrastive loss will be computed for the representation at each scale, and the final loss will be the average of all the scales. This design allows PLanTS to jointly capture fine-grained local patterns and broader state transitions. It is worth noting that the multi-granularity patching mechanism is only applied during the self-supervised learning phase to construct training views; in the testing phase, the pretrained encoder is used directly on raw sequences without patching.
>
> To illustrate this, we will add the following paragraph in section 3.2 Periodicity-aware multi-granularity patching:
> “Each granularity reflects a distinct temporal resolution. PLanTS encodes these patches independently through the latent-state and dynamic-transition encoders, compute a contrastive loss for each of them, then integrate the information by taking the average of all these losses. This hierarchical structure enables the model to capture both within-state variations and between-state transitions consistently across scales.”
>
> Response to weakness 2: The baselines are pretty old, new methods should be presented, such as CSL[1].
>
> We appreciate the reviewer for suggesting the new baseline. We have now evaluated CSL and incorporated its results into our comparison. Since CSL is primarily designed for classification tasks, we have included its performance in Table 1. Please refer to our response in Weakness 4 for the updated table. The revised results will also be reflected in the updated PDF.

---

> ### Author Response · Authors · 2025-11-20
>
> Response to weakness 3: Figure 2 introduces pretext tasks for TNC and TS2Vec. However the purpose for demonstrating them, e.g., for contrastive usage, is not introduced explicitly enough.
>
> We appreciate the reviewer for pointing out that there is not sufficient explanation for Figure 2(a). We introduced the goal of Figure 2(a) in its caption but didn’t provide sufficient discussions in the main text. We want to clarify that Figure 2(a) serves to compare different contrastive learning paradigms in multivariate time series (MTS). Specifically, TNC and TS2Vec are included to illustrate (1) window-based and (2) time-point–based contrastive formulations, respectively, both of which motivate our design of PLanTS. We will add the following paragraph at the beginning of section 3.2 to explicitly state this purpose:
>
> “Figure 2(a) compares contrastive learning paradigms from prior self-supervised learning methods with our proposed PLanTS framework. TNC formulates a fixed window-based contrastive task, defining temporally neighboring windows as positives and distant ones as negatives, while TS2Vec adopts a time-point–based contrastive formulation, encouraging contextual consistency at each timestamp. Both methods are hard contrastive methods and cannot adapt to the diverse periodicities. In contrast, PLanTS introduces a periodicity-aware, multi-granularity soft contrastive mechanism that leverages temporal similarity without relying on hard positive–negative sampling.”
>
> Response to weakness 4: Taking an average over performances on 30 UEA dataset is not a reasonable operation, because there is no statistical meaning. [1] Liang, Z., Zhang, J., Liang, C., Wang, H., Liang, Z., & Pan, L. (2023). A Shapelet-based Framework for Unsupervised Multivariate Time Series Representation Learning. Proc. VLDB Endow., 17, 386-399.
>
> We appreciate the reviewer for pointing out the limitation in one of our evaluation metrics. To strengthen the statistical validity of our comparison, we additionally adopted the metric used in CSL. Specifically, for each baseline, we report the number of datasets in which PLanTS wins / ties / loses (W/T/L) in one-versus-one comparisons. We then applied the Wilcoxon rank test’s p-values to evaluate the statistical significance. For fairer comparison, we rerun all the baselines with the same hardware, environment, and experimental settings. The updated comparison table is shown below. As illustrated, PLanTS significantly outperforms all competing methods on the UEA datasets (p-value < 0.05 under most circumstances).  Full results table will be provided in our updated pdf.
>
> | **Method** | **Average Accuracy** | **Average Rank** | **Times Ranked 1st** | **Avg. Accuracy Diff (%)** | **W/T/L** | **Wilcoxon p-value** |
> |-----------|-----------------------|------------------|------------------------|--------------------------------------|-----------|------------------------|
> | PLanTS    | 0.72                  | 3.333            | 8                      | 0                                    | –         | –                      |
> | CLS       | 0.698                 | 4.867            | 7                      | 6.447                                | 18/0/12   | 0.089                  |
> | softclt   | 0.709                 | 4.481            | 5                      | 4.559                                | 16/1/10   | 0.033                  |
> | T-Rep     | 0.693                 | 4.667            | 4                      | 5.303                                | 18/5/7    | 0.036                  |
> | TS2vec    | 0.69                  | 5.1              | 3                      | 4.407                                | 21/0/9    | 0.003                  |
> | T-Loss    | 0.658                 | 4.833            | 5                      | 7.643                                | 17/4/9    | 0.009                  |
> | TNC       | 0.67                  | 5.767            | 2                      | 5.653                                | 23/3/4    | 0                      |
> | TS-TCC    | 0.668                 | 5.633            | 4                      | 6.49                                 | 20/2/8    | 0.001                  |
> | TST       | 0.617                 | 6.9              | 2                      | 10.877                               | 24/2/4    | 0                      |
> | DTW       | 0.65                  | 5.862            | 1                      | 9.241                                | 21/1/7    | 0.001                  |

---

### Meta-Review · Area_Chair_KhSz · 2026-01-10

**Summary:**

The paper proposes a self-supervised representation learning framework for multivariate time series (MTS) data, aiming to fill the gap that recent self-supervised learning (SSL) methods overlook the intrinsic periodic structure of MTS. It models latent states and their transitions using a periodicity-aware multi-granularity patching mechanism and a contrastive loss across multiple temporal resolutions. Experiments on multiple downstream tasks, including classification, forecasting, trajectory tracking, and anomaly detection, demonstrate the advantage of this framework over the baseline.

All four reviewers agree that the paper is marginally below the acceptance threshold, and two of them give the highest level of confidence. The reviewers raise concerns regarding the pretty old baselines, lack of salient analysis and insights, and missing related works like time series foundation models.

**Reviewer Concerns:**

There are two major concerns after the rebuttal:

1. Reviewer oiHR and rZwx raise concerns regarding the lack of salient analysis and insights, e.g., the motivations for designing the multi-granularity patching mechanism is unclear. The authors also acknowledge this in the rebuttal.

2. Reviewer tVjj questions why multi-scale modeling can address non-stationary challenges in MTS. The authors attribute non-stationarity in their rebuttal to temporal dependencies and feature correlations varying across different time horizons, and explain it from this perspective. However, I believe this attribution is not rigorous and requires more experimental evidence.

**Reviewer Scores:**

Reviewer oiHR: Rated the paper marginally below the acceptance threshold with confidence 5.  Although no explicit post-rebuttal update was provided, the rebuttal's explanation of Insights still lacked a rigorous analysis of the shortcomings of existing methods, thus necessitating the introduction of the designed module. The score would likely remain unchanged.

Reviewer tVjj: Rated the paper marginally below the acceptance threshold and the concern 2 was still not explained with experimental support in the author's rebuttal. The score would likely remain unchanged.

Reviewer rZwx: Rated the paper marginally below the acceptance threshold with confidence 5. The reviewer mentioned that more datasets except for ETTs are needed. Although the authors included Weather dataset in the rebuttal, but the traffic dataset which is commonly used in forecasting task was still missed. The score would likely remain unchanged.

Reviewer us21: Rated the paper marginally below the acceptance threshold, the rebuttal explained the reviewers' concerns about the next-transition prediction task in detail, so I think the score would increase slightly.

---

### Decision · Program_Chairs · 2026-01-26

Reject